# GuidedSampling: Steering LLMs Towards Diverse Candidate Solutions at Inference-Time

**Divij Handa**[1]    **Mihir Parmar**[1]*   **Aswin RRV**[1]    **Md Nayem Uddin**[1]
**Hamid Palangi**[2]    **Chitta Baral**[1]
[1]Arizona State University    [2] Google
{dhanda, mihirparmar, cbaral}@asu.edu

## Abstract

Repeated Sampling (RS) is a simple inference-time algorithm that has been shown to improve model performance on complex tasks. Although it is an effective way of scaling inference time, it often struggles to generate diverse solution candidates, frequently relying on the same underlying approach to solve the problem and thus producing redundant samples. To address this limitation, we propose a new inference algorithm, GuidedSampling, which decouples the exploration and generation phases during inference, increasing diversity of generated candidate solutions. The exploration phase identifies multiple concepts that can be utilized to solve the problem, while the generation phase applies a specific concept to provide final solution candidates. We first define the theoretical bounds of GuidedSampling and then empirically demonstrate that it improves the performance of base model at pass@50 by on an average $\sim 21.6\%$ across various benchmarks compared to RS. Furthermore, models trained on trajectories of GuidedSampling exhibit substantial performance improvements at pass@5 by on an average $\sim 9.7\%$, compared to models trained on traditional RS. Additionally, models trained with GuidedSampling increases the average number of concepts per instance $(1.67 \rightarrow 3.03)$, yielding a diverse set of candidates than traditional RS. [1]

## 1 Introduction

Recent advances in large language models (LLMs) have shown that scaling model size and training data can lead to increasingly capable systems across diverse domains, including mathematical reasoning, scientific analysis, and code generation (Kaplan et al., 2020). However, scaling models indefinitely is becoming increasingly infeasible due to the requirement of more data for training ever-larger models (Villalobos et al., 2024). As a result, a growing body of work has shifted focus to alternative ways of boosting performance—not by making models larger, but by making better use of available compute during inference (Hosseini et al., 2024; Kumar et al., 2024; Lightman et al.,

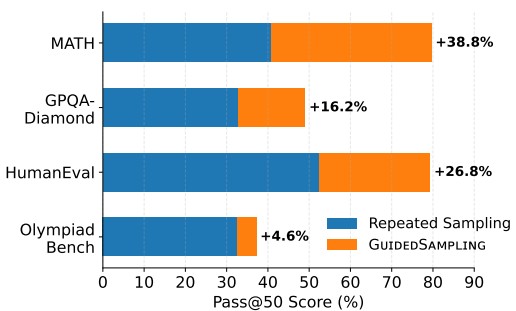

Figure 1: Pass@50 improvements with best performing base model using GuidedSampling.

2023; Brown et al., 2024). Several studies now suggest that allocating additional compute at inference time can lead to larger performance gains than spending that compute to train bigger models (Snell et al., 2024; Wu et al., 2024). This has led to a fundamental shift in improving the performance of inference-time algorithms (Muennighoff et al., 2025; Ghosal et al., 2025).

Recently, various inference-time algorithms have been proposed (Wang et al., 2022; Yao et al., 2023; Zhang et al., 2024). Among them, repeated sampling (RS) (Cobbe et al., 2021) is one of

---

*Now at Google

[1]The code and data is available at https://github.com/DivijH/sampling_inference

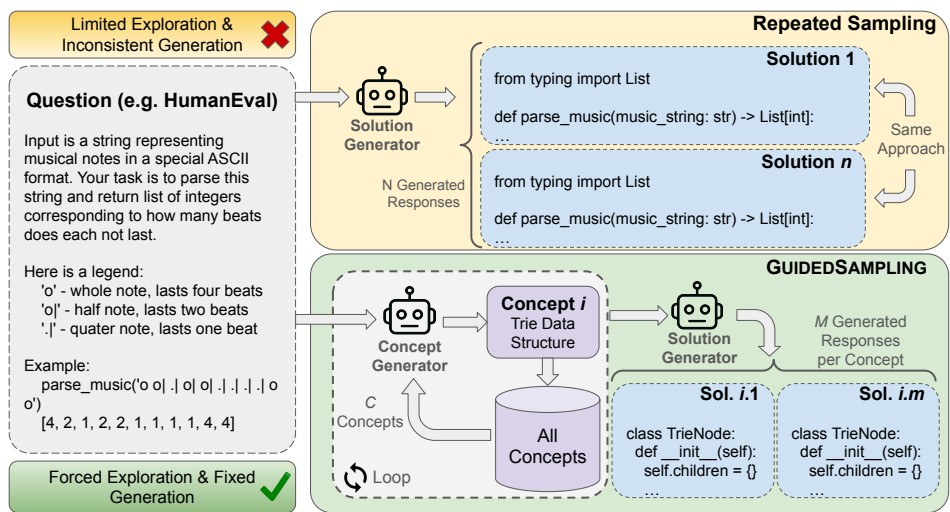

Figure 2: GUIDEDSAMPLING enhances exploration during inference by first generating a set of diverse concepts or theorems to guide subsequent generations of solutions. Unlike repeated sampling (RS), where the model generates the final solution, GUIDEDSAMPLING separates these phases.

the most widely used inference-time algorithms, where multiple outputs are sampled for the same input prompt. Traditional RS "*implicitly*" combines two phases: *exploration*, which we define as identifying the diverse theorems or concepts used in solving a given question, and *generation*, where the LLMs use the identified concept and try to generate several candidate solutions for the problem. However, despite its simplicity, traditional RS suffers from a lack of exploration (Brown et al., 2024), due to LLMs being traditionally trained to generate a **single** correct response for every input (Chow et al., 2024). This leads RS to generate solutions with the same underlying concepts rather than a thorough exploration of the solution space. To address this limitation, we propose inference-time algorithm, GUIDEDSAMPLING, designed to decouple the exploration of diverse concepts from the generation of final solutions. We define theoretical bounds for GUIDEDSAMPLING (§3.3), and then empirically demonstrate how training LLMs on such trajectories shows significant pass@k gains.

GUIDEDSAMPLING (Figure 2) first explicitly explores diverse concepts that can be used to solve a given question. For our experiments, we define concepts as the names of the theorems that can be utilized for solving questions (examples in Appendix C). In the second phase, these concepts guide the generation of complete candidate solutions. This decoupling is the key reason that GUIDED-SAMPLING enhances the diversity of solution candidates generated during inference, and also gives explicit control over exploration. As illustrated in Figure 1, our experiments on Llama-3.2-3B-Instruct (Grattafiori et al., 2024) and Qwen2.5-3B-Instruct (Yang et al., 2024) show an improvement at pass@50 on MATH for mathematical reasoning (Hendrycks et al., 2021), GPQA-Diamond for scientific reasoning (Rein et al., 2024), HumanEval for Python code generation (Chen et al., 2021), and OlympiadBench for complex mathematical and scientific reasoning (He et al., 2024). Further analysis by extracting the concept present in the candidate solutions generated by base models (§3.1) reveal that GUIDEDSAMPLING generates $17.63\%$ more diverse candidate solutions compared to RS.

For instance, consider a problem from MATH: "Find the maximum value of $\left[\frac{x-y}{x^4+y^4+6}\right]$ over all real numbers $x$ and $y$.". For this problem, we sample 1000 solutions using traditional RS and GUIDED-SAMPLING. Our detailed analysis of concepts extracted from these candidates shows that $892/1000$ uses the "*AM-GM inequality*" concept to solve the problem, consistently leading to the incorrect solution due to over-utilizing the same theorem. In contrast, only $77/1000$ candidates from GUID-EDSAMPLING use this theorem, dedicating the remaining compute to exploring other theorems such as "*Cauchy-Schwarz Inequality*", "*Trivial Inequality*", and "*Chebyshev's Inequality*".

Our other core contribution is to use GUIDEDSAMPLING to improve LLM post-training. We demonstrate that fine-tuning LLMs on trajectories generated by GUIDEDSAMPLING outperforms models trained on trajectories from traditional RS, Tree-of-Thought (Yao et al., 2023), and other

self-correction methods like Self-Taught Reasoner (STaR) (Zelikman et al., 2022). We generate diverse solution trajectories using GUIDEDSAMPLING on a random subset of 10k instances from OpenMathInstruct-2 (Toshniwal et al., 2024), a mathematical reasoning dataset. LLMs fine-tuned on this data exhibited a $3.43\% \uparrow$ in pass@5 on the MATH benchmark. These fine-tuned models also demonstrate improved generalization, with pass@5 gains on out-of-domain benchmarks, GPQA-Diamond ($6.17\% \uparrow$), HumanEval ($1.86\% \uparrow$), and OlympiadBench ($2.11\% \uparrow$) compared to the strongest baseline. In summary, GUIDEDSAMPLING facilitates future research towards exploring diversity at inference-time and can effectively synthesize exploration-aware data for post-training.

## 2 RELATED WORKS

**Inference-Time Strategies** Chain-of-thought (CoT) and its variants (Wei et al., 2022; Kojima et al., 2022) showed that guiding LLMs to produce intermediate reasoning steps during inference boosts performance on complex tasks such as mathematical reasoning. However, as reasoning chains become longer, CoT suffers from error propagation due to complex calculations (Chen et al., 2022). To mitigate this, new methods have been proposed, e.g., Self-Consistency, which samples multiple CoT and then selects the most consistent answer through majority voting (Wang et al., 2022). Building upon these ideas, better search algorithms, such as the tree-of-thought (ToT) (Yao et al., 2023), MCTS (Zhang et al., 2024), and REBASE (Wu et al., 2024), have been proposed, which enable LLMs to perform more deliberate problem-solving by exploring multiple reasoning paths in a tree structure. Several agentic systems (Parmar et al., 2025; Handa et al., 2025; Zhang et al., 2025; Kumbhar et al., 2025) have shown that performing multi-agent debate before generating a final solution improves performance. Furthermore, recent work (Muennighoff et al., 2025) has extended the 'thinking' of models by introducing special tokens such as "wait" to improve performance. Finally, Ghosal et al. (2025) has shown that simply sampling from a model repeatedly outperforms such approaches. In contrast to prior methods, GUIDEDSAMPLING generates a diverse set of samples with lower inference-time cost than tree search (Yao et al., 2023), while achieving greater diversity than standard sampling approaches. Parallel to our work, Wang et al. (2025) proposed *RandIdeaInjection*, which first generates a list of distinct ideas and then injects the generated list into the generation process to produce the final response. GUIDEDSAMPLING, on the other hand, works in an iterative loop of generating concepts, adding them individually to generate the final output.

**Synthetic Data w/ Inference-Time Algorithms** Recent works have explored leveraging inference strategies for generating high-quality synthetic training data and fine-tuning models to improve their performance. For instance, Self-Taught Reasoner (STaR) (Zelikman et al., 2022) is an iterative method where an LLM is prompted to generate CoT rationales; those rationales that lead to correct answers are then used as high-quality synthetic data to fine-tune the model, while those which lead to incorrect answers are passed back to model for refinement along with the correct final answer, effectively bootstrapping its reasoning abilities from a small initial set. Similarly, ReST$^{EM}$ (Singh et al., 2023), building on principles of reinforced self-training (ReST), employs an iterative Expectation-Maximization-like framework. It uses Best-of-N (BoN) sampling to generate multiple candidate solutions for problems and then refines the model by training on this synthetically generated data. Some works (Chow et al., 2024; Tang et al., 2025; Rrv et al., 2025) have developed reinforcement learning methods that directly optimize for pass@k metrics, leading to significant gains in reasoning and code generation. Other methods, such as multi-agent fine-tuning (Subramaniam et al., 2025), train diverse agent models through debate and voting, while Gui et al. (2024) introduced BoNBoN Alignment, distilling the BoN distribution into a single model. While these strategies improve pass@k, they often do not explicitly manage the trade-off between exploration and generation. In contrast, our proposed GUIDEDSAMPLING method introduces a structured exploration phase during training, explicitly balancing diversity and quality, and models fine-tuned with our trajectories outperform those trained using methods like BoN, STaR, or ToT.

## 3 GUIDEDSAMPLING

### 3.1 BACKGROUND

**Traditional RS** Repeated Sampling (RS) is a simple strategy to increase the inference-time performance of a model by generating multiple samples from the model's output distribution. Let

$X = \{x_1, x_2, \ldots, x_N\}$ be a set of input queries. For each input $x \in X$, we draw $k$ independent samples from the model-defined conditional distribution $p_\theta(y \mid x)$, i.e.,

$$y_i^{(x)} \sim p_\theta(y \mid x), \quad \text{for } i = 1, \ldots, k$$

This process effectively scales the model's inference-time compute linearly with $k$. The theoretical appeal of RS lies in its potential to achieve complete coverage of the output space as $k \to \infty$. For any output $y^*$ such that $p_\theta(y^* \mid x) > 0$, the probability that it's sampled at least once after $k$ samples:

$$P_k = 1 - (1 - p_\theta(y^* \mid x))^k$$

This quantity monotonically increases with $k$ and asymptotically approaches 1. Thus, under the assumption that all valid outputs are assigned non-zero probability by the model, unlimited sampling ensures that the target output will be generated at least once. This has led to several works adopting RS to generate solutions (Wang et al., 2022; Rozière et al., 2023; Li et al., 2022). Of course, unlimited sampling is impractical. The value of RS lies in whether increased sampling leads to improved output quality within a feasible compute budget. Also, several state that the lack of diversity in these generated responses is the key limitation of scaling RS (Brown et al., 2024; Wang et al., 2025).

**Diversity Analysis** To quantify the lack of diversity in RS, we use Qwen2.5-32B-Instruct (Yang et al., 2024) to extract the core concept or theorem from each solution. We present the prompt for concept extraction in Appendix B.2. We find that solutions sampled using RS tend to rely heavily on a few underlying concepts to solve the problem, even with increasing the amount of compute. For instance, Llama-3.2-3B-Instruct used an average of 2.75 different concepts while solving code generation questions from the HumanEval benchmark, even with 100 candidate solutions. Figure 3 represents the distribution of the number of questions for how many concepts are generated for a fixed budget of 100 responses. We observe that in 64% of the questions, fewer than three concepts were used to solve the questions, with 36.4% just using one concept.

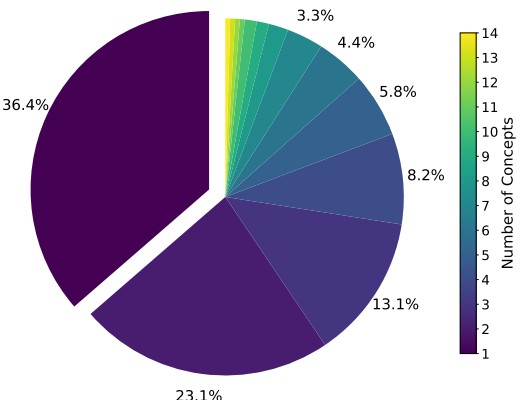

Figure 3: Distribution of the number of concepts used by Llama-3.2-3B-Instruct for 100 candidates. 37% of the questions are attempted with just one concept, while less than 36% of the questions have more than two concepts.

**Tree-of-Thoughts (ToT)** ToT represents a more sophisticated strategy for enhancing model performance in complex problem-solving tasks by explicitly exploring multiple reasoning paths (Yao et al., 2023). ToT guides a language model to generate a tree of "thoughts", where each thought $t_i$ is a coherent sequence of text representing an intermediate step towards a solution. The model generates multiple candidate thoughts $T_j = \{t_1^{(j)}, t_2^{(j)}, \ldots, t_m^{(j)}\}$ from a parent thought $t_p$. Each of these candidate thoughts is then evaluated, often by the LLM itself or a separate verifier, $V(t_i^{(j)} \mid P, t_p)$, to assess its promise. Search algorithms like Breadth-First Search (BFS) or Depth-First Search (DFS) are employed to navigate this tree, allowing the model to look ahead, backtrack if a path seems unpromising, and explore different lines of reasoning (Long, 2023). The theoretical strength of ToT lies in its potential to systematically explore a vast solution space, thereby increasing the likelihood of finding a correct or high-quality solution, especially for tasks where simpler methods like Chain of Thought (CoT) might falter due to their linear, single-path reasoning. This structured exploration aims to address issues like the lack of diversity in generated paths by deliberately generating and considering varied intermediate steps. However, this explicit generation and evaluation of numerous thought branches make tree-of-thought computationally intensive, with costs scaling with the number of candidates explored at each step ($m$) and the depth of the tree (Yao et al., 2023).

While ToT solves the lack of diversity observed in RS (Appendix D), it is significantly more computational as explicit evaluation of each intermediate thought generated at every step of the tree's

expansion is required. To mitigate both the lack of diversity in the solutions and less computational cost, we propose GUIDEDSAMPLING, which we elaborate on in the following sections.

## 3.2 OUR PROPOSED APPROACH

Our proposed inference algorithm, GUIDEDSAMPLING, improves the diversity by separating exploration and generation into two distinct phases. This separation allows for finer control over the diversity of concepts that can be used to solve a problem, an aspect previous approaches like traditional RS fall short of. Moreover, our method explores the concepts just once in the beginning, which leads to better efficiency than the tree-of-thought strategy. Figure 2 highlights the differences between our strategy and RS. We describe these two phases of our strategy in detail below:

**Exploration Phase**   The goal of the Exploration Phase is to discover a diverse set of high-level ideas, concepts, or theorems that could guide the solution of a given question. We start with a dataset or a set of questions denoted by $X$, from which we sample a specific question $x \in X$ to work on. Given this question $x$ and an LLM parameterized by $\theta$, we aim to identify a set of relevant concepts that could support downstream reasoning or problem-solving, denoted as $\mathcal{C} = \{c_1, c_2, ..., c_K\}$. The process of constructing $\mathcal{C}$ is iterative: the $k$-th concept is generated by conditioning on the original question $x$ and all previously generated concepts $c_1, \ldots, c_{k-1}$. Formally, this sampling process is expressed as:

$$c_k \sim p_\theta(\cdot \mid x, c_{1:(k-1)})$$

This iterative conditioning mechanism promotes diversity among the concepts, encouraging the model to explore different areas of the solution space rather than repeating similar concepts. The algorithm continues until either $K$ concepts have been generated or the model determines that no more useful concepts can be produced—allowing for early stopping. The prompts used for exploration are presented in Appendix B.1, and some concept examples are illustrated in Appendix C.

**Generation Phase**   Once the set of candidate concepts $\mathcal{C} = c_1, c_2, \ldots, c_K$ has been established during the Exploration Phase, the Generation Phase uses these concepts to produce concrete solutions. For each concept $c_k \in \mathcal{C}$, we generate $M$ potential solutions. These solutions are sampled from the LLM, conditioned on both the original question $x$ and the specific concept $c_k$:

$$\mathcal{S}_k = \left\{ s_k^{(m)} \sim p_\theta(s \mid x, c_k) \right\}_{m=1}^{M}$$

Each completion $s_k^{(m)}$ represents a full solution that uses the guidance provided by $c_k$. The full set of candidate solutions is thus $\mathcal{S} = \bigcup_{k=1}^{K} \mathcal{S}_k$.

This structured sampling strategy leverages the earlier exploration to guide the solutions more effectively. Instead of relying on unguided or purely random repeated sampling, the model systematically explores multiple reasoning trajectories guided by diverse high-level concepts or theorems. This enhances the diversity of candidate solutions, increasing the likelihood that at least one solution will be correct. We formally define the GUIDEDSAMPLING algorithm in Algorithm 1.

## 3.3 THEORETICAL BOUNDS FOR GUIDEDSAMPLING

**Definition 1** (Notation). *Let $x$ be the input prompt and $y^*$ be a correct final solution. Let $\pi_{base}(y \mid x)$ be the base model's conditional probability of generating solution $y$ directly from $x$. In the GUIDEDSAMPLING framework, we define:*

- *$c$: An intermediate concept or theorem.*

- *$\mathcal{C}_r$: The set of "relevant" concepts that contain a valid concept pointing towards the correct reasoning path $y^*$.*

- *$\pi_{concept}(c \mid x)$: The probability of generating concept $c$ from prompt $x$.*

- *$\pi_{solution}(y \mid x, c)$: Probability of generating solution $y$ given the prompt $x$ and concept $c$.*

- *$\mathcal{I}(y; c \mid x)$: sample-wise mutual information between $y$ and $c$ conditional on $x$. This represents the additional information contributed by the concept $c$ in predicting $y$.*

By intuition, solving a question becomes easier if we know a good problem-appropriate "hint" for a question. To elaborate on the performance bounds of GUIDEDSAMPLING, we make the following assumption:

**Assumption 1.** *For any "relevant" concept $c \in \mathcal{C}_r$, conditioning on it strictly increases the probability of generating a correct solution $y^*$. That is, there exists an amplification factor $k_c > 1$ such that:*

$$\pi_{base}(y^* \mid x, c) \geq k_c \cdot \pi_{base}(y^* \mid x) \tag{1}$$

The above assumption is based on the intuition that $\mathcal{I}(y; c \mid x) > 0$, i.e., any "relevant" concept strictly increases the probability of generating the correct final response. For "irrelevant" concepts ($c \notin \mathcal{C}_r$), the assumption doesn't hold. We also bridge the intuition to above assumption in Appendix A.1. Following the above assumption, we now state our main theorem:

**Theorem 1.** *Let $P_{RS}(y^* \mid x)$ be the probability of generating a correct solution through Repeated Sampling and $P_{GS}(y^* \mid x)$ be the probability of generating a correct solution through GUIDED-SAMPLING. Under Assumption 1, $P_{GS}(y^* \mid x) > P_{RS}(y^* \mid x)$ iff the following condition holds:*

$$(k_{min} \cdot P(\mathcal{C}_r \mid x) - 1) \cdot P_{RS}(y^* \mid x) + \sum_{c \notin \mathcal{C}_r} \pi_{concept}(c \mid x) \cdot \pi_{base}(y^* \mid x, c) > 0 \tag{2}$$

*where $P(\mathcal{C}_r \mid x) = \sum_{c \in \mathcal{C}_r} \pi_{concept}(c \mid x)$ is the probability of generating a relevant concept, and $k_{min} > 1$ is the amplification factor in accordance with the above assumption.*

The condition derived in Theorem 1 provides a formal basis for when GUIDEDSAMPLING outperforms RS. We detail the proof in Appendix A.2. In practice, this condition is satisfied if one or more of the following pathways hold:

**Recovery from Irrelevant Concepts** If the second term, $\sum_{c \notin \mathcal{C}_r} \pi_{concept}(c \mid x) \cdot \pi_{base}(y^* \mid x, c)$, is sufficiently large. This corresponds to the scenario where the model generates a flawed or "irrelevant" concept but still manages to produce the correct solution, $y^*$. While this is possible, we observe empirically that it is a rare event. We detail one such case study in Appendix G. Therefore, for GUIDEDSAMPLING to be reliably superior, the following condition is more critical.

**Sufficient Concept Coverage** If first term, $(k_{min} \cdot P(\mathcal{C}_r \mid x) - 1) \cdot P_{RS}(y^* \mid x) > 0$. Since the second term is a probability distribution and will always remain $\geq 0$, for the overall sum in equation 2 to be positive, the first term should be positive. This holds when $P(\mathcal{C}_r \mid x) > 1/k_{min}$. This can be achieved either when the underlying model's probability of generating relevant concepts is high ($P(\mathcal{C}_r \mid x) \gg 0$), or when conditioning on a relevant concept provides a significant probabilistic advantage for generating the correct solution compared to direct generation ($k_{min} \gg 1$). We empirically observe both of these to be true for most cases in our study, but some models may lack this ability on certain tasks (e.g., Qwen2.5-3B-Instruct on code generation).

## 3.4 POST-TRAINING USING GUIDEDSAMPLING

Synthetic data has become an increasingly effective tool for enhancing the reasoning capabilities of LLMs (Gupta et al., 2023; Mitra et al., 2024; Chaudhary et al., 2023). In particular, inference-time algorithms are valuable for generating such data when the correctness of the final solution can be programmatically verified (Zelikman et al., 2022; Arora & Zanette, 2025; Shao et al., 2024). We demonstrate that GUIDEDSAMPLING can serve not only as an effective inference-time strategy but also as a powerful synthetic data generation mechanism.

Let $x$ denote an input question, and $\mathcal{C} = \{c_1, \ldots, c_K\}$ be the diverse set of concepts generated for $x$ using exploration phase of GUIDEDSAMPLING. For each concept $c_k \in \mathcal{C}$, we sample a solution $s \sim \mathcal{S}$. We define two distinct settings for constructing synthetic training pairs $(x, y)$:

1. **Final-Answer Only (FA):** In this setting, we discard the generated concept and only use the final verified response $s$ as the target output. This encourages the model to learn mappings from problem statements directly to correct answers, i.e. $(x, y) = (x, s)$. The corresponding training objective is the standard fine-tuning loss:

$$\mathcal{L}_{FA} = -\mathbb{E}_{(x,s) \sim \mathcal{D}_{FA}} [\log P_\theta(s \mid x)]$$

where $\mathcal{D}_{\text{FA}}$ is the dataset constructed under the FA regime and $P_\theta$ is the model's conditional distribution parameterized by $\theta$.

2. **Concept-Augmented Answer (CAA):** In the CAA setting, we construct an enriched target sequence that includes both the conceptual diversity and the final answer. Specifically, we concatenate the concepts $\mathcal{C}$ with one selected solution $s$ to form the training target:

$$(x, y) = (x, \texttt{concat}(\mathcal{C}, s))$$

This setting encourages the model to internalize multiple reasoning strategies before committing to one concrete solution path. The training objective becomes:

$$\mathcal{L}_{\text{CAA}} = -\mathbb{E}_{(x,\mathcal{C},s)\sim\mathcal{D}_{\text{CAA}}}\left[\log P_\theta(y \mid x)\right]$$

where $\mathcal{D}_{\text{CAA}}$ is the dataset constructed under the CAA regime. The prompt for CAA is provided in Appendix B.3.

## 4 EXPERIMENT SETUP

**Baselines**  We showcase GUIDEDSAMPLING against Repeated Sampling (RS) to showcase the better pass@k performance. For training, we compare models trained using Self-Taught Reasoner (STaR) (Zelikman et al., 2022), RS (Brown et al., 2024), and Tree-of-Thought (Yao et al., 2023).

**Dataset**  We use test sets of MATH (for mathematical reasoning) (Hendrycks et al., 2021), GPQA-Diamond (scientific reasoning) (Rein et al., 2024), HumanEval (code generation) (Chen et al., 2021), and OlympiadBench (mathematical and scientific reasoning) (He et al., 2024) to measure the effectiveness of GUIDEDSAMPLING. For training the models, we first randomly select 10k samples from the training set of OpenMathInstruct-2 (Toshniwal et al., 2024), math reasoning dataset. We then create reasoning chains using corresponding inference strategies and select the reasoning chains with correct final answer since ground truth is available to create corresponding training sets. We detail the fine-tuning setup in Appendix F.

**Models and Metrics**  We evaluate two open-source LLMs in our main study – Llama-3.2-3B-Instruct (Grattafiori et al., 2024) and Qwen2.5-3B-Instruct (Yang et al., 2024). We generate $n = 100$ responses using all models and report values until $k = 50$. For finetuned models, we generate $n = 10$ responses and report values until $k = 5$. Since our experiments involve generating up to 100 responses, we also perform a limited study of other models in Appendix E. To assess the performance, we use the pass@k metric, which is defined as the expected maximum reward obtained from the $k$ sampled responses out of $n$, where $c$ are correct candidates. Formally, it is defined as:

$$\text{pass@}k \;=\; \mathbb{E}\left[1 - \frac{\binom{n-c}{k}}{\binom{n}{k}}\right]$$

## 5 RESULTS AND DISCUSSION

**GUIDEDSAMPLING pass@k performance**  As shown in Figure 4, GUIDEDSAMPLING significantly outperforms RS across the majority of models and benchmark combinations. As an edge case, only one combination of Qwen2.5-3B-Instruct and HumanEval shows degradation in performance due to weak concept generation. Averaging across all models, we observe pass@50 improvements of 21.8% on MATH, 11.87% on GPQA-Diamond, 11.28% on HumanEval, and 3.08% on OlympiadBench. These results highlight that structured exploration enables more effective use of limited compute. However, the gains from GUIDEDSAMPLING are not uniform across all tasks and models. While Qwen2.5-3B-Instruct achieves strong improvements on MATH, its performance on HumanEval worsens compared to traditional RS. Upon closer analysis, this drop stems from Qwen's limited ability to generate diverse concepts for coding during the exploration phase. As mentioned in §3.3, a weaker probability of generating good concepts, $P(C_r \mid x)$, results in lower performance of GUIDEDSAMPLING. On average, Qwen produces only 1.13 distinct concepts per HumanEval problem, indicating that nearly all sampled solutions are guided by the same idea. This lack of diversity not only fails to leverage the core strengths of GUIDEDSAMPLING but can also dilute the

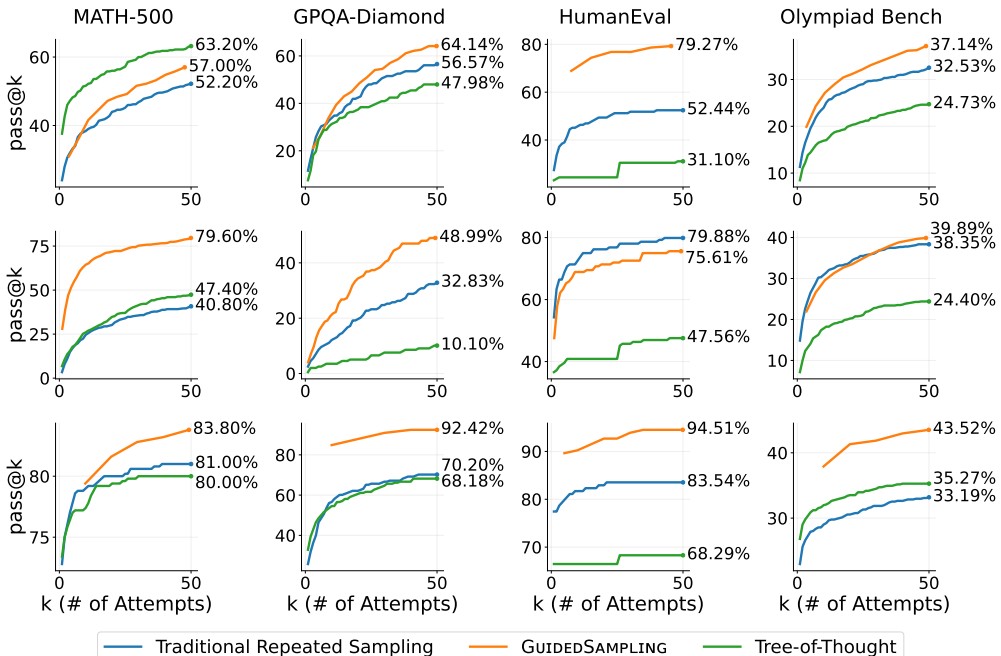

Figure 4: GUIDEDSAMPLING forces exploration during inference-time, resulting in $16.01\%$ average pass@k improvement compared to repeated sampling. We observe an average improvement of $21.8\%$ on MATH, $11.87\%$ on GPQA-Diamond, $11.28\%$ on HumanEval, and $3.08\%$ on OlympiadBench. For GUIDEDSAMPLING, we choose the optimal value of $K$ (from Fig. 5) that maximizes the performance. **First row**: Llama-3.2-3B-Instruct, **Second row**: Qwen2.5-3B-Instruct. **Third row**: Gemma-3-27b-it

model's effectiveness by forcing the model to follow a particular concept. In contrast, Llama-3.2-3B-Instruct generates 7.58 unique concepts on average on HumanEval, enabling richer exploration and stronger performance. These results underscore that the successful application of GUIDEDSAMPLING depends critically on the model's ability to generate varied and relevant high-level ideas. To validate whether the observed drop is due to poor concept generation or Qwen's inability to generate the correct solution from the concept, we use the concepts generated by Llama-3.2-3B-Instruct for generating the final answer. Using a stronger concept generator yields a pass@50 performance of 83.53%, a 3.65% improvement from RS. The smaller gains of Qwen on OlympiadBench can be attributed to the benchmark's high difficulty (olympiad-level problems) combined with the relatively small model size (3B). Nevertheless, GUIDEDSAMPLING still yields measurable improvements.

The higher performance of GUIDEDSAMPLING is due to $K$ concepts being generated. In practice, this value is far lower than the number of samples generated (100 in our case). Moreover, as the compute increases (increasing $k$ for pass@k), we observe that the performance gap between Repeated Sampling and GUIDEDSAMPLING increases in most cases (Fig. 4), suggesting that GUIDEDSAMPLING benefits more with increased compute. This leads us to believe that when computational resources are sufficient, a small overhead of sequential calls for generating concepts might be a beneficial tradeoff for better performance.

**Diversity in GUIDEDSAMPLING** To measure the diversity of candidate solutions, we use Qwen2.5-32B-Instruct (Yang et al., 2024) to extract the core concept or theorem. We then compute the number of distinct concepts generated. On average, RS produces 3.54, 6.72, 2.66, and 3.25 distinct concepts on MATH, GPQA-Diamond, HumanEval, and OlympiadBench, respectively. GUIDEDSAMPLING produces 3.66, 7.66, 3.87, and 3.81 distinct concepts, improving the diversity by an average of 17.63%. We also found the diversity gains from GUIDEDSAMPLING are model-specific. We find that Llama-3.2-3B-Instruct generates $3.7\times$ more unique concepts on average compared to Qwen2.5-3B-Instruct, with this gap ranging from $2.82\times$ on GPQA-Diamond to $5.12\times$ on HumanEval. This suggests that model architecture and pretraining influence the capacity for generating novel reasoning strategies. We show examples of generated concepts in Appendix C.

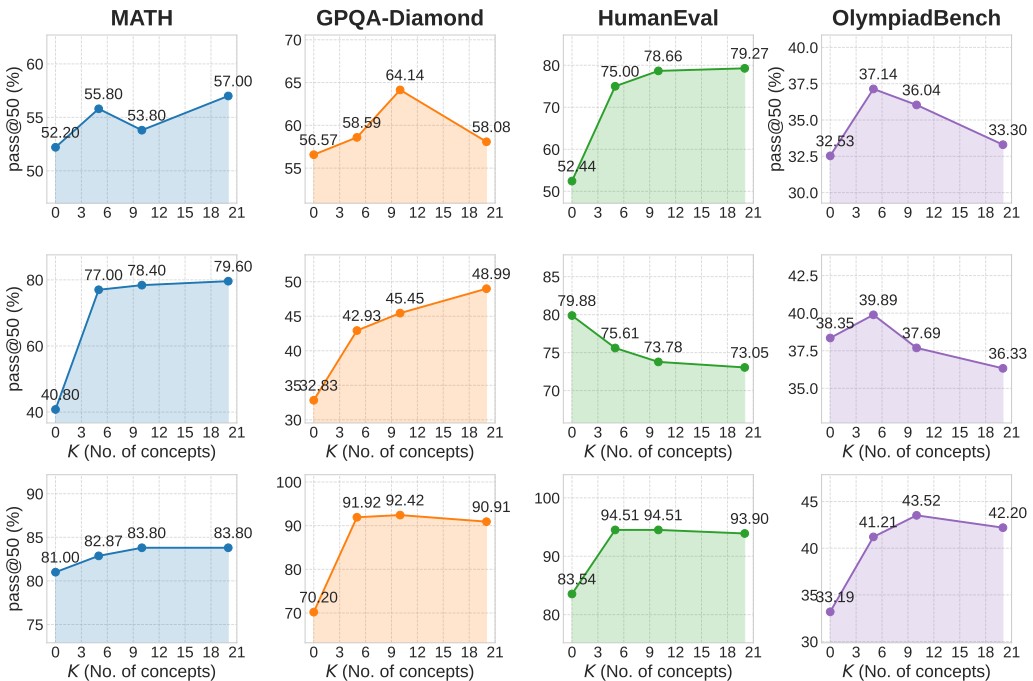

Figure 5: Pass@50 performance variation with different exploration (number of concepts $K$) and generation (samples per idea $M$) compute allocations, given a fixed total compute of 100 calls ($M = 100/K$). Increasing exploration initially helps, but performance declines when the generation budget per idea becomes too small. At $K = 0$, GUIDEDSAMPLING becomes traditional RS. **First row**: Llama-3.2-3B-Instruct, **Second row**: Qwen2.5-3B-Instruct. **Third row**: Gemma-3-27b-it

**Trade-off between Exploration and Generation**  A key design choice in GUIDEDSAMPLING is the allocation of the limited inference compute budget $IC$ between the exploration phase (number of concepts $K$) and the generation phase (number of samples $M$ per concept, where $M = IC/K$). The number of distinct concepts $K$ directly controls this trade-off: a larger $K$ encourages broader exploration of different approaches, but consequently reduces the compute available for generating solutions using each approach (i.e., smaller $M$). Conversely, a smaller $K$ allows for more generations using fewer concepts. As demonstrated in Fig. 5, increasing exploration by increasing $K$ initially boosts performance in most cases by uncovering more diverse, potentially successful strategies. However, beyond an optimal point, performance may decline as the generation budget $M$ for each concept becomes insufficient to thoroughly develop any single approach.

**Performance of Earlier vs Later Concepts**  During the concept generation phase of GUIDED-SAMPLING, concepts are generated iteratively. To determine the contribution of the $k$-th concept across all questions that produced at least $k$ concepts, we analyzed all models and benchmarks mentioned in §4, which contains a total of 1772 questions. We observe a minor decline from $k = 1$ to $5$ ($19.8\% \rightarrow 16.2\%$). This observation suggests that earlier concepts suggested by the concept generator are better than later ones. However, for concepts with index $k \geq 6$ (i.e., when more exploration is needed), a higher performance variance due to a sharp decrease in coverage is observed. E.g., only 72 out of 1772 questions reach $k = 9$, meaning there are fewer samples with $k \geq 6$ concepts. This results in variations in performance, with higher performance being observed for many such cases (e.g., 52.05% performance for $k = 14$, due to just 23 instances). Thus, although the earlier concepts are beneficial, later ones ($k \geq 6$) also contribute to increasing performance, but for a small number of instances that require significant exploration. Hence, for the overall success of GuidedSampling, even the later ones are also important, but the earlier ones play a major role. Individual performance values are provided in Appendix I.

**Dependence of GUIDEDSAMPLING on Well-defined Concepts**   Theorem 1 states that for a "relevant" concept $c$, conditioning on $c$ increases the probability of generating the correct solution. However, in domains such as commonsense reasoning, which involve more imprecise, vague, and uncertain knowledge, defining such concepts is difficult. Hence, the condition stated in Theorem 1, i.e., $P(C_r|x) >> 0$, might not be satisfied. Applying GUIDEDSAMPLING to Qwen2.5-3B-Instruct on CommonSenseQA (Talmor et al., 2019), a commonsense benchmark. The model is prompted to generate a general idea that could help solve the question (not a task-specific concept, since those are lacking in the commonsense domain). On such domains, GUIDEDSAMPLING underperforms against Repeated Sampling by 3.28% (pass@50). Based on this, we believe that GUIDEDSAMPLING has a better chance of succeeding when concepts can be formulated efficiently. More details in Appendix E.

**Final Answer Selection via Majority Voting**   To select a final solution after sampling multiple times, we use the majority voting technique, where the most common solution is selected as the final answer. GUIDEDSAMPLING achieves an average accuracy of 35.87% compared to Repeated Sampling (32.80%) and Tree-of-Thought (26.26%). Detailed accuracies in Appendix E.3.

**Finetuning models on GUIDEDSAMPLING trajectories**   Models fine-tuned on data synthesized via GUIDEDSAMPLING significantly outperform those trained using data from other inference-time algorithms, as illustrated in Table 1. Notably, when the models are asked to produce more responses (pass@5), a bigger improvement in performance is observed. On average, the CAA setting yields 7.13% pass@5 improvements compared to the RS, while FA shows 5.64% pass@5 improvements against RS. Models trained using trajectories from Tree-of-Thought, another explorative strategy, performed better than RS as well, showing a 4.37% improvement, but still underperformed when compared against GUIDEDSAMPLING: FA (1.45%) and CAA (2.76%).

Table 1: Performance of Llama-3.2-3B-Instruct trained using different synthetic data creation strategies. FA: Using just the final answer for training the model. CAA: Using both the concepts and the corresponding final solution to create the training data.

| Method | MATH | | GPQA-Diamond | | HumanEval | | OlympiadBench | |
|---|---|---|---|---|---|---|---|---|
| | pass@1 | pass@5 | pass@1 | pass@5 | pass@1 | pass@5 | pass@1 | pass@5 |
| Base Model | 24.00% | 33.20% | 11.62% | 28.28% | 27.44% | 39.02% | **11.32%** | 19.56% |
| RS | 37.62% | 44.78% | 18.13% | 40.08% | 52.13% | 55.78% | 6.42% | 10.83% |
| STaR | 36.60% | 46.23% | 16.61% | 38.41% | 52.13% | 57.35% | 5.82% | 10.62% |
| ToT | **40.40%** | 56.63% | 16.77% | 44.44% | 35.73% | 49.51% | 9.19% | 18.36% |
| FA (Ours) | 29.88% | 47.98% | **20.20%** | **50.61%** | 48.17% | 55.95% | 11.21% | 20.21% |
| CAA (Ours) | 38.00% | **60.06%** | 15.66% | 40.23% | **53.05%** | **59.21%** | 10.76% | **20.47%** |

**Diversity of Solutions by Finetuned Models**   We extract the core concept or theory used in the candidate solutions and observe that diversity increases from 1.67 (RS) to 2.58 (FA) and 3.03 (CAA). Surprisingly, the largest diversity gain occurs on GPQA-Diamond rather than MATH, indicating that diversity learned through training on mathematical reasoning data can transfer to other domains. This highlights the generalizability of the GUIDEDSAMPLING framework across domains.

## 6   CONCLUSIONS

We propose a new inference-time algorithm, GUIDEDSAMPLING, that forces exploration of candidate solutions over repeated sampling. The paper demonstrates how performance varies with shifting compute between the exploration of diverse concepts and the generation of final solutions and shows pass@50 improvements of up to 34.6%. Furthermore, fine-tuning LLMs on trajectories generated by GUIDEDSAMPLING significantly boosts performance on mathematical reasoning and shows generalizability to other domains like scientific reasoning and code generation.

## LIMITATIONS & FUTURE WORK

While our method is successful in improving the diversity of solutions generated by LLMs, it represents an early step in this area and has some limitations, including but not limited to the following:

**Limited model coverage**   While our evaluation spans three open-source models and a limited study on GPT-4o-mini and Phi-4-mini-instruct, applying GUIDEDSAMPLING to proprietary models (e.g., GPT-5, Gemini-2.5-Pro) remains unexplored due to high inference costs and lack of training access. Extending the method to state-of-the-art models is an important direction for future work.

**Exploration cost vs. effectiveness trade-off**   Although our method improves diversity, the optimal balance between the number of concepts ($K$) and samples per concept ($M$) under a fixed compute budget remains task-specific. Developing adaptive strategies for this trade-off is a promising area.

**Generality across domains**   Our work demonstrates promising results in mathematical, scientific, and code generation domains. However, further evaluation is needed to understand how well GUIDEDSAMPLING generalizes to more diverse domains such as legal reasoning, medical, or discovery.

**Concept generation quality**   The success of GUIDEDSAMPLING depends on the quality and diversity of the generated concepts. Although some "irrelevant" concepts can help (Appendix G), investigating techniques to improve/verify the relevance of these concepts (e.g., through external tools or feedback mechanisms) can enhance overall effectiveness.

**Better Verifier**   GUIDEDSAMPLING has an exploration phase, which forces the model to explore multiple concepts, increasing diversity. This can lead to multiple final solutions. While this increases pass@k, building a robust verifier that can select a final solution, even if it is in the minority, remains a challenging future task.

## REPRODUBILITY STATEMENT

To ensure the reproducibility of our results, we release the source code and data through our GitHub repo `https://github.com/DivijH/sampling_inference`. We also note that LLMs are inherently probabilistic in nature, and some results may vary upon each run. We hope our code and data aid in future research.

## ETHICS STATEMENT

In accordance with ICLR policy, we disclose that AI assistants, specifically Grammarly for grammar correction and ChatGPT for sentence restructuring and paraphrasing, were utilized during the preparation of this manuscript. The authors have reviewed, edited, and take full responsibility for all final content presented in this paper.

## ACKNOWLEDGEMENT

We thank the anonymous reviewers for their constructive suggestions. We extend our gratitude to the Research Computing (RC) and Enterprise Technology at ASU for providing computing resources for experiments.

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

# A  THEORETICAL PROOFS

## A.1  FROM INTUITION TO ASSUMPTION 1

Assumption 1 stems from the intuition that any "relevant" concept helps in answering a given question, i.e., the concept adds more information which is useful. This is represented as sample-wise information between $y$ and $c$ conditioned on $x$.

$$\mathcal{I}(y; c \mid x) > 0$$
$$\log \pi_{base}(y^* \mid x, c) - \log \pi_{base}(y^* \mid x) > 0$$
$$\log \pi_{base}(y^* \mid x, c) > \log \pi_{base}(y^* \mid x)$$
$$\pi_{base}(y^* \mid x, c) \geq k_c \cdot \pi_{base}(y^* \mid x) \tag{3}$$

This is the stated assumption 1.

## A.2  PROOF OF THEOREM 1

*Proof.* The probability of generating a correct solution via Repeated Sampling is given by:

$$P_{RS}(y^* \mid x) = \pi_{base}(y^* \mid x) \tag{4}$$

For GUIDEDSAMPLING, the probability of generating a correct solution:

$$P_{GS}(y^* \mid x) = \sum_c \pi_{concept}(c \mid x) \cdot \pi_{base}(y^* \mid x, c) \tag{5}$$

We can partition the sum based on whether the concept is in the set of relevant concepts, $\mathcal{C}_r$:

$$P_{GS}(y^* \mid x) = \sum_{c \in \mathcal{C}_r} \pi_{concept}(c \mid x) \cdot \pi_{base}(y^* \mid x, c) + \sum_{c \notin \mathcal{C}_r} \pi_{concept}(c \mid x) \cdot \pi_{base}(y^* \mid x, c) \tag{6}$$

Let's analyze the first term. By Assumption 1, for any informative concept $c \in \mathcal{C}_{inf}$, we have $\pi_{base}(y^* \mid x, c) = k_c \cdot \pi_{base}(y^* \mid x)$ where $k_c > 1$. Intuitively, since relevant concepts tend to be informative, we can say that for any relevant concept $c \in \mathcal{C}_r$, we have $\pi_{base}(y^* \mid x, c) = k_c \cdot \pi_{base}(y^* \mid x)$. Let $k_{min} = \min_{c \in \mathcal{C}_r} k_c$. It follows that $k_{min} > 1$. We can therefore lower-bound the first term:

$$\sum_{c \in \mathcal{C}_r} \pi_{concept}(c \mid x) \cdot \pi_{base}(y^* \mid x, c) \geq \sum_{c \in \mathcal{C}_r} \pi_{concept}(c \mid x) \cdot (k_{min} \cdot \pi_{base}(y^* \mid x)) \tag{7}$$

$$= k_{min} \cdot \pi_{base}(y^* \mid x) \sum_{c \in \mathcal{C}_r} \pi_{concept}(c \mid x) \tag{8}$$

$$= k_{min} \cdot P_{RS}(y^* \mid x) \cdot P(\mathcal{C}_r \mid x) \tag{9}$$

where $P(\mathcal{C}_r \mid x)$ is the total probability of sampling a valid concept.

Substituting this back into our expression for $P_{GS}(y^* \mid x)$ (Eq. 5), we get:

$$P_{GS}(y^* \mid x) \geq k_{min} \cdot P_{RS}(y^* \mid x) \cdot P(\mathcal{C}_r \mid x) + \sum_{c \notin \mathcal{C}_r} \pi_{concept}(c \mid x) \cdot \pi_{base}(y^* \mid x, c) \tag{10}$$

For GUIDEDSAMPLING to be superior to repeated sampling, we require $P_{GS}(y^*|x) > P_{RS}(y^*|x)$. This inequality holds if:

$$k_{min} \cdot P_{RS}(y^* \mid x) \cdot P(\mathcal{C}_r \mid x) + \sum_{c \notin \mathcal{C}_r} \pi_{concept}(c \mid x) \cdot \pi_{base}(y^* \mid x, c) > P_{RS}(y^* \mid x) \tag{11}$$

Rearranging the terms yields the condition stated in the theorem:

$$(k_{min} \cdot P(\mathcal{C}_r \mid x) - 1) \cdot P_{RS}(y^* \mid x) + \sum_{c \notin \mathcal{C}_r} \pi_{concept}(c \mid x) \cdot \pi_{base}(y^* \mid x, c) > 0 \tag{12}$$

$\square$

# B PROMPTS USED IN OUR STUDY

## B.1 EXPLORATION PROMPTS

### B.1.1 MATH

The following prompts were used for GUIDEDSAMPLING for the MATH (Hendrycks et al., 2021) benchmark.

---

**MATH Initial Concept Generation**

You are an expert mathematician. You will be presented with a mathematical question and your task is to identify and state one single, specific theorem or fundamental concept that is most relevant and useful for solving the problem.

QUESTION:
{ele['question']}

Provide only the name of the theorem or concept, or a concise statement of the principle, that is most directly applicable to solving this problem. Do not attempt to solve the original problem. Only provide the theorem or concept.

---

**MATH Subsequent Concept Generation**

You are an expert mathematician. You will be presented with a mathematical question and a list of theorems and concepts that have already been proposed as potentially useful for solving the problem. Your task is to provide a *new* and *different* theorem or concept that is most relevant and useful for solving the problem.

QUESTION:
{ele['question']}

EXISTING CONCEPTS:
{ideas_text}

Provide only the name of the theorem or concept, or a concise statement of the principle, that is most directly applicable to solving this problem. Do not attempt to solve the original problem. Only provide the theorem or concept. If no new, distinct, and useful theorem or concept can be identified, respond with "No additional concepts found."

---

### B.1.2 GPQA-DIAMOND

The following prompts were used for GUIDEDSAMPLING for the GPQA-Diamond (Rein et al., 2024) benchmark.

---

**GPQA-Diamond Initial Concept Generation**

You are an expert scientist and problem solver. You will be presented with a complex, graduate-level science question and your task is to identify and state one single, specific theorem or fundamental concept that is most relevant and useful for solving the problem.

QUESTION:
{ele['question']}{options}

Provide only the name of the theorem or concept, or a concise statement of the principle, that is most directly applicable to solving this problem. Do not attempt to solve the original problem. Only provide the theorem or concept.

---

---

**GPQA-Diamond Subsequent Concept Generation**

You are an expert scientist and problem solver. You will be presented with a complex, graduate-level science question and a list of theorems and concepts that have already been proposed as potentially useful for solving the problem. Your task is to provide a *new* and *different* theorem or concept that is most relevant and useful for solving the problem.

QUESTION:
{ele['question']}{options}

EXISTING CONCEPTS:
{ideas_text}

Provide only the name of the theorem or concept, or a concise statement of the principle, that is most directly applicable to solving this problem. Do not attempt to solve the original problem. Only provide the theorem or concept. If no new, distinct, and useful theorem or concept can be identified, respond with "No additional concepts found."

---

### B.1.3 HUMANEVAL

The following prompts were used for GUIDEDSAMPLING for the HumanEval (Chen et al., 2021) benchmark.

---

**HumanEval Initial Concept Generation**

You are an expert python programmer. You will be presented with a programming question and your task is to identify and state one single, specific concept that is most relevant and useful for solving the problem.

QUESTION:
{ele['question']}

Provide only the name or short description of the concept, that is most directly applicable to solving this problem. Do not attempt to solve the original question. Only provide the concept.

---

**HumanEval Subsequent Concept Generation**

You are an expert python programmer. You will be presented with a programming question and a list of concepts that have already been proposed as potentially useful for solving the question. Your task is to provide a *new* and *different* concept that is most relevant and useful for solving the question.

QUESTION:
{ele['question']}

EXISTING CONCEPTS:
{ideas_text}

Provide only the name or the short description of the concept, that is most directly applicable to solving this problem. Do not attempt to solve the original question. Only provide the concept. If no new, distinct, and useful concept can be identified, respond with "No additional concepts found."

---

### B.1.4 OLYMPIADBENCH

The following prompts were used for GUIDEDSAMPLING for the OlympiadBench (He et al., 2024) benchmark.

---

**OlympiadBench Initial Concept Generation**

You are an expert scientist. You will be presented with a question and your task is to identify and state one single, specific theorem or concept that is most relevant and useful for solving the problem.

QUESTION:
`{ele['question']}`
Provide only the name of the theorem or concept that is most directly applicable to solving this problem. Do not attempt to solve the original problem. Only provide a single theorem or concept.

---

**OlympiadBench Subsequent Concept Generation**

You are an expert scientist. You will be presented with a question and a list of theorems and concepts that have already been proposed as potentially useful for solving the problem. Your task is to provide a single **new** and **different** theorem or concept that is most relevant and useful for solving the problem. Do not elaborate on the theorem or concept. If no new, distinct, and useful theorem or concept can be identified, respond with "No additional concepts found."

QUESTION:
`{ele['question']}`

EXISTING CONCEPTS:
`{ideas_text}`

Provide only the name of a single new and different theorem or concept that is most directly applicable to solving this problem. Do not attempt to solve the original problem. If no new, distinct, and useful theorem or concept can be identified, respond with "No additional concepts found."

---

### B.2 CONCEPT EXTRACTION PROMPT

---

**Concept Extraction Prompt**

You are ConceptTagger, an expert that maps a worked-out solution (chain-of-thought or final answer) to the most specific mathematical or logical concept that makes the solution possible.

Task: For every input consisting of a reasoning explanation (a step-by-step solution, scratch-work, or short justification):
1. Read the explanation.
2. Decide which single mathematical concept, theorem, or canonical formula is essential for the solution.
3. Output that concept's standard name—nothing else.

---

Choose the narrowest concept that still covers the whole solution.
• Good: "Pythagorean Theorem" (precise).
• Bad: "Geometry" (too broad).
If two or more concepts appear, pick the one without which the problem cannot be solved (typically the first pivotal step).

Here are two examples:

### Example 1
Problem: A right triangle has legs of lengths 5 cm and 12 cm. What is the length of the hypotenuse?
Step-by-step solution:
Step 1: Recognize this is a right triangle → apply the Pythagorean Theorem.
Step 2: hypotenuse = $\sqrt{(5^2 + 12^2)} = \sqrt{(25 + 144)} = \sqrt{169} = 13cm$
Concept Used: Pythagorean Theorem

### Example 2
Problem: What is the area of a rectangle with a length of 9 meters and width of 4 meters?
Step-by-step solution:
Step 1: Identify the shape as a rectangle.
Step 2: Use the area formula: Area = length × width = 9 × 4 = 36 m²
Concept Used: Area of Rectangle

Formatting Rules:
Output exactly one line with the concept name.
Use Title Case and the singular form (e.g., "Least Common Multiple", not "LCMs").
No extra punctuation, explanation, or line breaks.

## B.3 CAA PROMPT

**CAA Data**

I have a few ideas to solve this problem.
a) {Concept 1}
⋮
k) {Concept k}

To solve the problem I will use the idea i) {Concept i}:

{Step by step solution}

**Final Answer**
{Final Answer}

## C CONCEPT EXAMPLES

In this section, we detail some examples from each benchmark and the concepts generated by Repeated Sampling and GUIDEDSAMPLING. We extract the concepts using Qwen2.5-32B-Instruct.

## C.1 CONCEPT EXAMPLES IN MATH

For the following question from the MATH benchmark, Table 2 displays the generated concepts related to the above question.

> Convert the point $(0, 3)$ in rectangular coordinates to polar coordinates. Enter your answer in the form $(r, \theta)$, where $r > 0$ and $0 \le \theta < 2\pi$.

Table 2: Concepts generated via Repeated Sampling and GUIDEDSAMPLING on a MATH instance.

| Repeated Sampling | GUIDEDSAMPLING |
|---|---|
| Polar Coordinates Conversion | Distance Formula |
| | Inverse Circular Function |
| | Trigonometric Identity |
| | Circular Function |
| | Pythagorean Theorem |
| | Polar Coordinate Transformation |

## C.2 CONCEPT EXAMPLES IN GPQA-DIAMOND

For the following question from the GPQA-Diamond benchmark, Table 3 displays the generated concepts related to the above question.

> Two quantum states with energies E1 and E2 have a lifetime of $10^{-9}$ sec and $10^{-8}$ sec, respectively. We want to clearly distinguish these two energy levels. Which one of the following options could be their energy difference so that they can be clearly resolved?

Table 3: Concepts generated via Repeated Sampling and GUIDEDSAMPLING on a GPQA-Diamond instance.

| Repeated Sampling | GUIDEDSAMPLING |
|---|---|
| Heisenberg Uncertainty Principle | Heisenberg Uncertainty Principle |
| | Stark Shift |
| | Quantum Rabi Frequency |

## C.3 CONCEPT EXAMPLES IN HUMANEVAL

For the following question from the HumanEval benchmark, Table 4 displays the generated concepts related to the above question.

```
from typing import List

def separate_paren_groups(paren_string: str) -> List[str]:
    """ Input to this function is a string containing multiple
    groups of nested parentheses. Your goal is to separate
    those group into separate strings and return the list of
    those.
    Separate groups are balanced (each open brace is properly
    closed) and not nested within each other
    Ignore any spaces in the input string.
    >>> separate_paren_groups('( ) (( )) (( )( ))')
    ['()', '(())', '(()())']
    """
```

## C.4 CONCEPT EXAMPLES IN OLYMPIADBENCH

For the following question from the OlympiadBench benchmark, Table 5 displays the generated concepts related to the above question.

Table 4: Concepts generated via Repeated Sampling and GUIDEDSAMPLING on a HumanEval instance.

| Repeated Sampling | GUIDEDSAMPLING |
|---|---|
| Stack | Graph-Based Approach with a Stack |
| Parentheses Matching | Balanced Parentheses Tree Construction |
| Stack Manipulation Space Ignoring | Recursive Descent Parsing |
| | Prefix Tree Traversal |
| | Dynamic Programming with Memoization |
| | Level Order Traversal with a Queue |
| | Suffix Tree Construction with a Stack |
| | Counter-Based Approach with a Stack |
| | Kruskal's Algorithm with a Union-Find Data Structure |
| | Nested Set Algorithm |

> Xenia and Sergey play the following game. Xenia thinks of a positive integer $N$ not exceeding 5000. Then she fixes 20 distinct positive integers $a_1, a_2, \ldots, a_{20}$ such that, for each $k = 1, 2, \ldots, 20$, the numbers $N$ and $a_k$ are congruent modulo $k$. By a move, Sergey tells Xenia a set $S$ of positive integers not exceeding 20 , and she tells him back the set $\{a_k : k \in S\}$ without spelling out which number corresponds to which index. How many moves does Sergey need to determine for sure the number Xenia thought of?

Table 5: Concepts generated via Repeated Sampling and GUIDEDSAMPLING on a GPQA-Diamond instance.

| Repeated Sampling | GUIDEDSAMPLING |
|---|---|
| Chinese Remainder Theorem | Pigeonhole Principle |
| Inclusion-Exclusion Principle | Chebyshev's Postulate |
| Pick's Theorem | Erdős-Szekeres Lemma |
| | Sperner's Lemma |
| | Dirichlet's Box Principle |
| | Hadamard's Lemma |
| | König's Theorem |

## D   DIVERSITY ANALYSIS OF INFERENCE-TIME ALGORITHMS

Here we detail the diversity analysis of Repeated Sampling (RS), Tree-of-Thought (ToT), and GUIDEDSAMPLING. We use Qwen-2.5-32B-Instruct to extract the concepts used in each candidate solution. We observe an average of 4.04 concepts in RS, while in GUIDEDSAMPLING, we observe 4.75 different concepts, with less compute budget. With ToT, on the other hand, we observe 4.25 average concepts.

## E   MORE RESULTS USING GUIDEDSAMPLING

### E.1   RESULTS FOR MORE LLMS

In this section, we showcase some results on additional models. As mentioned in §4, we generate 100 candidate solutions for each instance. We provide results on Phi-4-mini-instruct (Abouelenin et al., 2025), GPT-4o-mini (Hurst et al., 2024), and Gemma-3-27b-it (Team et al., 2025). Due to limited resource constraints, we limit the proprietary model to just the MATH (Hendrycks et al., 2021) benchmark. Table 6 and 7 show the pass@50 results for these models along with the observed diversity as extracted by Qwen-3.2-32B-Instruct (Yang et al., 2024). Diversity is measured by the average number of concepts for each instance.

Table 6: pass@50 performance of GPT-4o-mini and Phi-4-mini-instruct on MATH, along with diversity of concepts observed in candidate solutions. RS: Repeated Sampling, GS: GUIDEDSAMPLING

| Model | Repeated Sampling | GUIDEDSAMPLING | Diversity in RS | Diversity in GS |
|---|---|---|---|---|
| GPT-4o-mini | 85.71% | 90.00% | 3.2 | 5.0 |
| Phi-4-mini-instruct | 71.80% | 80.80% | 2.1 | 3.4 |

Table 7: pass@50 performance of Gemma-3-27b-it

| Benchmark | Repeated Sampling | GUIDEDSAMPLING |
|---|---|---|
| MATH | 81.00% | 83.80% |
| GPQA-Diamond | 70.20% | 92.42% |
| HumanEval | 83.54% | 94.51% |
| OlympiadBench | 33.19% | 43.52% |

## E.2 RESULTS ON COMMONSENSEQA

Results for Qwen2.5-3B-Instruct on CommonSenseQA are reported in Table 8. The prompts used don't specify a task-specific definition of concepts. Prompts are as follows:

---

**CommonSenseQA Initial Concept Generation**

You are a helpful assistant. Your task is to state a concept that is relevant and useful for answering the question.

QUESTION:
`{ele['question']}`

Provide the concept that is most directly applicable to answering the question. Do not answer the original question.

---

**CommonSenseQA Subsequent Concept Generation**

You are a helpful assistant. You will be presented with a question and a list of concepts that have already been proposed as potentially useful for answering the question. Your task is to provide a *new* and *different* concept that is relevant and useful for answering the question.

QUESTION:
`{ele['question']}`

EXISTING CONCEPTS:
`{ideas_text}`

Provide the concept that is most directly applicable to answering the question. Do not answer the original question. If no new, distinct, and useful concept can be identified, respond with "No additional concepts found."

---

Table 8: pass@50 performance of Qwen2.5-3B-Instruct on CommonSenseQA. RS: Repeated Sampling, GS: GUIDEDSAMPLING

| Repeated Sampling | GUIDEDSAMPLING |
|---|---|
| 98.94% | 95.66% |

## E.3  MAJORITY VOTING RESULTS

Table 9 shows the overall accuracies of Majority Voting applied on top of Repeated Sampling, GUID-EDSAMPLING, and Tree-of-thought. Out of the 8 different settings, GUIDEDSAMPLING achieves better accuracy in 4 of them, and a higher average performance as well.

Table 9: Accuracy of models on benchmarks using majority voting.

| Benchmark | Model | Repeated Sampling | GUIDEDSAMPLING | Tree-of-thought |
|---|---|---|---|---|
| MATH | Llama-3.2-3B-Instruct | 50.40% | 43.40% | 45.80% |
| GPQA-Diamond | Llama-3.2-3B-Instruct | 23.23% | 23.23% | 19.19% |
| HumanEval | Llama-3.2-3B-Instruct | 20.12% | 45.12% | 25.61% |
| OlympiadBench | Llama-3.2-3B-Instruct | 17.47% | 18.35% | 12.75% |
| MATH | Qwen2.5-3B-Instruct | 51.20% | 64.20% | 45.40% |
| GPQA-Diamond | Qwen2.5-3B-Instruct | 20.71% | 20.20% | 7.07% |
| HumanEval | Qwen2.5-3B-Instruct | 56.71% | 50.61% | 39.02% |
| OlympiadBench | Qwen2.5-3B-Instruct | 22.53% | 21.87% | 15.27% |
| Average | - | 32.80% | 35.87% | 26.26% |

## F  FINETUNING SETUP

Here we define the hyperparameters that we used for fine-tuning defined in Section 3.4.

All the models were trained on $4 \times$ A100 GPUs, with a learning rate of $5e^{-5}$ and 3 epochs. Batch size and Gradient accumulation steps were 2, and fp16 was used for all experiments. 20% of the data was split for evaluation (random seed as 21), and the checkpoint with the lowest evaluation loss was considered for reporting the results.

To determine whether the model trained using CAA trajectories experiences any collapse, we use one common observation: a collapsed model can repeat tokens indefinitely without generating an end-of-sequence token during inference. While model collapse has been studied to occur for several reasons (Shumailov et al., 2024; Gerstgrasser et al., 2024), checking for repeated tokens can indicate whether collapse happens or not.

To validate this, we run the base model and the model trained on CAA trajectories on HumanEval with 10 candidate solutions and check the *"finish_reason"* [2] after generation. Both the base model and the model trained using CAA trajectories return with the finish reason of *"stop"*, indicating that the model produced the end-of-sequence token.

Moreover, our evaluation, as described in §4, requires the model to generate responses in a particular format (refer to Appendix B.3 for the prompt). Since we did see improvements in pass@k values, we can be assured that the model followed the format closely.

Based on these two insights, we can say with some evidence that no model collapse occurred while training with CAA trajectories.

## G  CASE STUDY: CORRECT SOLUTION FROM "IRRELEVANT" CONCEPTS

As noted in §3.3, one approach for GUIDEDSAMPLING to outperform repeated sampling is through "Recovery from Irrelevant Concepts". While our empirical results (Figure 4) show that generating concepts boosts pass@k performance, we observe "irrelevant" concepts generated in some instances. Below, we detail one such example where the concept is unrelated at first glance, but leads to the correct solution.

---

[2] `https://docs.vllm.ai/en/latest/api/vllm/outputs/#vllm.outputs.CompletionOutput.finish_reason`

> **Question from HumanEval**
>
> ```
> from typing import List
> def has_close_elements(numbers: List[float], threshold: float)
> -> bool:
>     '''
>     Check if in given list of numbers, are any two numbers
>     closer to each other than given threshold.
>     >>> has_close_elements([1.0, 2.0, 3.0], 0.5)
>     False
>     >>> has_close_elements([1.0, 2.8, 3.0, 4.0, 5.0, 2.0], 0.3)
>     True
>     '''
> ```

To solve this problem, the model suggests 4 different concepts – Range Tree, K-D Tree or Octree, Bloom Filter, and Sorting. The simplest and most straightforward way to solve this problem is by using a sorting algorithm. While Range Tree and Octree also lead to correct solutions, they are more complex in nature. Bloom Filter, on the other hand, is **not** a correct concept for this problem. However, we still see some correct solutions leading from this concept. One such solution is given below:

> **Response from Llama-3.2-3B-Instruct**
>
> ```
> from typing import List
>
> def has_close_elements(numbers: List[float], threshold: float)
> -> bool:
>     # Create a set to store the numbers we've seen so far
>     seen = set()
>
>     # Iterate over the list of numbers
>     for num in numbers:
>         # For each number, check if there's a number in the set
>         # that's closer to it for seen_num in seen:
>             # If we find a number that's closer to the current
>             # number than the threshold
>             if abs(num - seen_num) < threshold:
>                 # Return True, because we've found two numbers
>                 # that are close to each other
>                 return True
>
>         # If we haven't found a close number yet, add the
>         # current number to the set
>         seen.add(num)
>
>     # If we've iterated over the entire list and haven't found
>     # two close numbers, return False
>     return False
> ```

The above solution is inspired by the Bloom Filter concept, but does not use the concept as-is, making it an approach that has never been observed in RS. Instances like this make GUIDEDSAMPLING more explorative than RS.

## H  ALGORITHM FOR GUIDEDSAMPLING

Here, we present the algorithm 1 for GUIDEDSAMPLING:

---

**Algorithm 1** GUIDEDSAMPLING

1: **Input:** Question prompt $x$, LLM $p_\theta$, maximum number of ideas $K$, completions per idea $M$
2: **Output:** Set of candidate solutions $\mathcal{S}$
3:
4: *// Exploration Phase*
5: $\mathcal{C} \leftarrow \emptyset$                                                                      ▷ Initialize set of concepts
6: $k \leftarrow 1$
7: **while** $k \leq K$ **do**
8:     $c_k \sim p_\theta(\cdot \mid x, c_1, \ldots, c_{k-1})$                                              ▷ Sample concept
9:     **if** $c_k = \text{None}$ **then**                                       ▷ Model indicates no more useful concepts
10:         **break**
11:     **end if**
12:     $\mathcal{C} \leftarrow \mathcal{C} \cup \{c_k\}$
13:     $k \leftarrow k + 1$
14: **end while**
15:
16: *// Generation Phase*
17: $\mathcal{S} \leftarrow \emptyset$                                                                      ▷ Initialize set of solutions
18: **for** each concept $c_k \in \mathcal{C}$ **do**
19:     $\mathcal{S}_k \leftarrow \emptyset$                                                                ▷ Initialize solutions for current concept
20:     **for** $m = 1$ **to** $M$ **do**
21:         Sample solution $s_k^{(m)} \sim p_\theta(\cdot \mid x, c_k)$                                    ▷ Generate solution based on concept
22:         $\mathcal{S}_k \leftarrow \mathcal{S}_k \cup \{s_k^{(m)}\}$
23:     **end for**
24:     $\mathcal{S} \leftarrow \mathcal{S} \cup \mathcal{S}_k$
25: **end for**
26: **return** $\mathcal{S}$

---

# I  PERFORMANCE VARIATION FOR $k$-TH CONCEPT

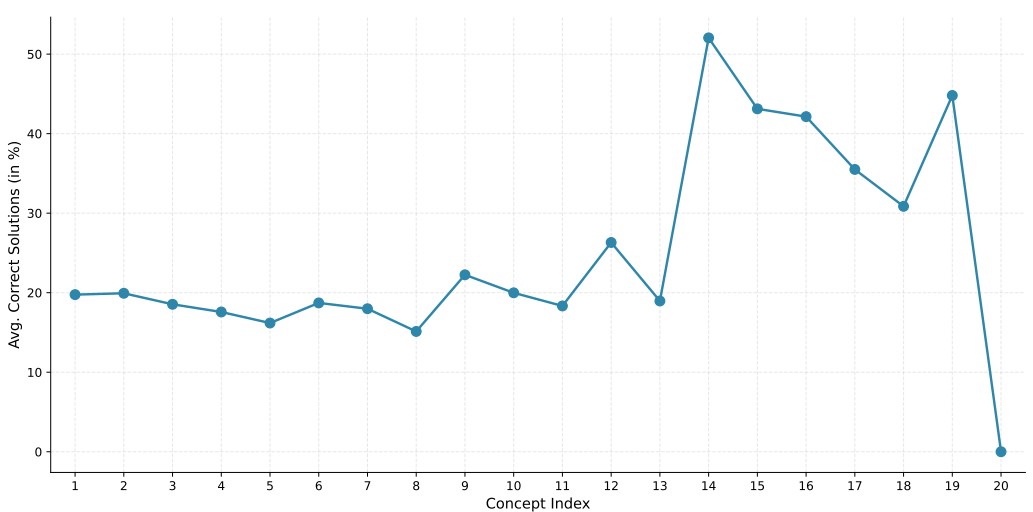

Figure 6: Pass@50 performance variation for $k$-th concept averaged across all benchmarks and mdoels mentioned in §4.

Here we detail the individual performance of the $k$-th concept across every model and benchmark. Fig. 6 illustrates the performance for every concept. As discussed in §5. Since later concepts have fewer instances, we see a huge variation in performance. Table 10 shows the detailed performance and number of instances for all concepts.

Table 10: pass@50 performance and the number of instances for the $k$-th concept generated in GUIDEDSAMPLING across all benchmarks and models, resulting in a total of 1772 instances.

| Concept Index | Avg. Correct Solutions (in %) | Number of Instances |
|:---:|:---:|:---:|
| 1 | 19.76 % | 1772 (100.00%) |
| 2 | 19.93 % | 1646 (92.89%) |
| 3 | 18.54 % | 1473 (83.13%) |
| 4 | 17.57 % | 1193 (67.33%) |
| 5 | 16.19 % | 819 (46.22%) |
| 6 | 18.71 % | 178 (10.05%) |
| 7 | 17.98 % | 126 (7.11%) |
| 8 | 15.13 % | 89 (5.02%) |
| 9 | 22.25 % | 72 (4.06%) |
| 10 | 19.98 % | 59 (3.33%) |
| 11 | 18.34 % | 47 (2.65%) |
| 12 | 26.31 % | 39 (2.20%) |
| 13 | 18.96 % | 28 (1.58%) |
| 14 | 52.05 % | 23 (1.30%) |
| 15 | 43.12 % | 16 (0.90%) |
| 16 | 42.14 % | 14 (0.79%) |
| 17 | 35.50 % | 8 (0.45%) |
| 18 | 30.86 % | 7 (0.40%) |
| 19 | 44.80 % | 5 (0.28%) |
| 20 | 0.00 % | 1 (0.06%) |

## J   LATENCY OF INFERENCE-TIME ALGORITHMS

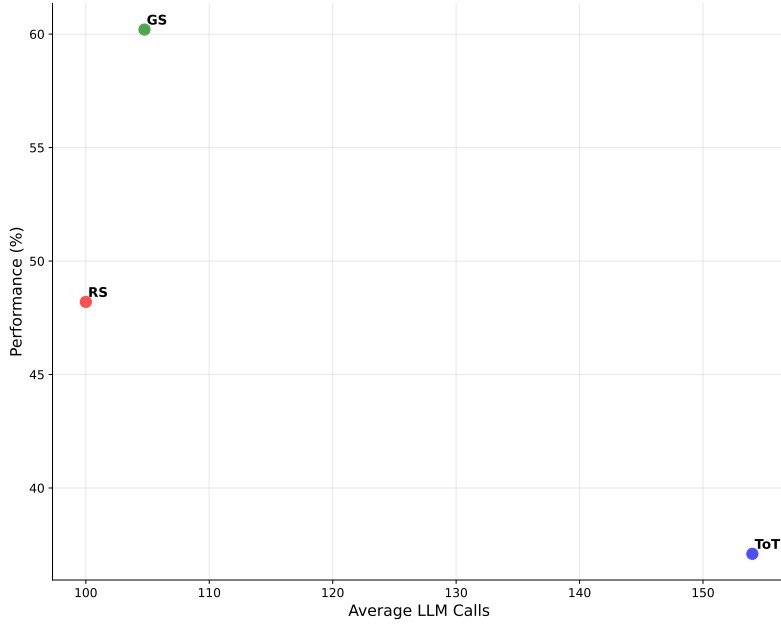

Figure 7: Pass@50 performance against the number of LLM calls for different inference-time algorithms averaged across all models and benchmarks.

Figure 7 shows the relationship between the number of LLM calls and pass@50 performance for Repeated Sampling (RS), GUIDEDSAMPLING (GS), and Tree-of-Thought (ToT). All results are averaged across all models and benchmarks. We found that GUIDEDSAMPLING (pass@50=60.2 with 104.75 calls) outperforms both Repeated Sampling (pass@50=48.2 with 100 calls) and Tree-of-Thought (pass@50=37.1 with 154 calls), while being more efficient than Tree-of-Thought.

