# OpenReview forum: "GuidedSampling: Steering LLMs Towards Diverse Candidate Solutions at Inference-Time"
_ICLR.cc/2026/Conference — ICLR 2026 Poster_

### Official Review · Reviewer_xAGJ · 2025-10-27

**Soundness:** 3
**Presentation:** 3
**Contribution:** 3
**Rating:** 6
**Confidence:** 3

**Summary:**

The paper proposes GuidedSampling, a novel inference-time algorithm designed to address the limited diversity of solutions generated by standard Repeated Sampling (RS). The core idea is to decouple inference into two phases:
1. Exploration: Generate a diverse set of high-level concepts (e.g., theorems, programming paradigms) relevant to solving the input problem.
2. Generation: For each concept, generate multiple candidate solutions conditioned on that concept.

The authors provide a theoretical analysis showing conditions under which GuidedSampling outperforms RS (Theorem 1). Empirically, they demonstrate clear pass@50 improvements over RS across four benchmarks. Additionally, they show that fine-tuning models on GuidedSampling-generated trajectories yields clear pass@5 gains and increases solution diversity.

**Strengths:**

1. While multi-path reasoning exists (e.g., ToT), GuidedSampling's two-phase design with iterative concept prompting shows better performance.

2. The paper is well-organized and easy-to-follow. It is supported with theoretical results.

**Weaknesses:**

1. The paper distinguishes itself from previous works via iterative concept prompting and theoretical analysis, but the high-level idea of separating idea generation from solution generation is not entirely unprecedented.

2. Theorem 1 assumes concepts amplify correctness probability (Assumption 1). However, irrelevant/misleading concepts could harm performance.

3. It seems necessary to design task-specific definitions of concepts, which limits generalization.

**Questions:**

1. Compared to the Tree-of-Thought (ToT) algorithm, GuidedSampling incurs lower computational overhead. However, ToT—being more expensive—should theoretically yield better performance, especially since GuidedSampling's approach appears subsumable within ToT’s framework (e.g., treating each concept as a root thought). Why then does ToT underperform relative to GuidedSampling in the experiments? Moreover, given the difference in computational cost, a direct comparison of inference-time overhead (e.g., total LLM calls, latency, or FLOPs) between the two methods is essential to contextualize their trade-offs.

2. How would GuidedSampling adapt to non-STEM tasks (e.g., commonsense reasoning, dialogue)? What would “concepts” represent in such settings?

---

> ### Author Response · Authors · 2025-11-23
>
> We thank the reviewer for their constructive feedback. Below, we address each point and specify the changes we made to the manuscript.
>
> ## Weaknesses
>
> **1.** We would like to point out that our focus in this work is to propose an inference-time algorithm that solves the diversity bottleneck observed in Repeated Sampling. We agree that, at a high level, the decoupling of concept and solution generation has been explored in the past, and we cite some relevant works in Related Works (Section 2). Our intent is not to claim this split for the first time, but to contribute a simple, inference-time algorithm with theoretical and empirical evidence. We summarize the novelty of our work below:
>
> a) Iterative, concept‑by‑concept prompting with early stopping
>
> b) A single control knob that splits a fixed inference compute budget $IC$ into $K$ concepts and $M = IC / K$ solutions per concept
>
> c) A lower bound for GuidedSampling (Theorem 1) that ties performance improvement to concept coverage and per‑concept uplift
>
> d) Training models on trajectories of GuidedSampling to internalize concept diversity
>
> **2.** The intuition behind Assumption 1 is that solving a question becomes easier if we know a good problem‑appropriate “hint” in the form of a concept. Irrelevant/misleading concepts contribute **no requirement** under Assumption 1. Thank you for your comment, and to clarify this further, we have added the following remark for Assumption 1:
>
> We only refer to “relevant” concepts ($c∈C_r$), i.e., concepts that are not irrelevant or misleading to the problem at hand. If a concept is misapplied or conflicts with the problem, it simply lies in $c∉C_r$. Our analysis separates these cases in Eq. (2): the “Recovery from Irrelevant Concepts” term can be small, zero, or occasionally positive (Appendix G provides a concrete case study where an “irrelevant” idea still yields a correct solution).
>
> **3.** In our work, we defined “concepts” as the names of the theorems or ideas for domains containing mathematical, scientific, or coding problems. As mentioned in Theorem 1, our framework requires a concept $c$ such that conditioning on $c$ increases the probability of generating the correct solution. In domains such as commonsense reasoning, which involve more imprecise, vague, and uncertain knowledge, defining such concepts is difficult. Hence, the condition stated in Theorem 1, i.e., $P(C_r|x) \gg 0$, might not be satisfied. To verify this, during the rebuttal period, we apply GuidedSampling to Qwen2.5-3B-Instruct on CommonSenseQA, a commonsense benchmark. We asked the model to generate a general idea that could help solve the question (not a task-specific concept, since those are lacking in the commonsense domain). We observed that GuidedSampling underperforms against Repeated Sampling by 3.28% (pass@50) (98.94% RS vs 95.66% GS). Thank you for pointing this out. We acknowledge this as a limitation of GuidedSampling. Based on this case study, we believe that GuidedSampling has a better chance of succeeding when concepts can be formulated efficiently. We have added this study and its corresponding prompts to Appendix E and referenced it in the main results (Section 5) for more insight.
>
> ## Questions
>
> **1.** We agree that ToT has more inference-time compute compared to GuidedSampling and should, in theory, perform better. However, ToT spends substantially more compute on intermediate thoughts and their evaluations than on producing diverse, fully‑worked candidate solutions. Furthermore, verification of intermediate steps can be noisy for 3B‑scale models, which prune promising but uncommon approaches [1,2,3]. GuidedSampling, by contrast, invests a tiny exploration budget to enumerate short, high‑level concepts and then spends the vast majority of compute on full solutions conditioned on each concept. Concretely, in our analysis, ToT produced fewer distinct concepts than GS at a higher cost (4.25 vs. 4.75 average concepts), and ToT‑generated synthetic data yielded smaller post‑training gains than GuidedSampling (Table 1). We have also included the total LLM calls as a measure for the cost of each algorithm in Appendix J.
>
> **2.** Please refer to Weakness 3.
>
> [1] Wang, Yidong, et al. "TrustJudge: Inconsistencies of LLM-as-a-Judge and How to Alleviate Them." arXiv preprint arXiv:2509.21117 (2025).
>
> [2] Zhang, Zhenru, et al. "The lessons of developing process reward models in mathematical reasoning." arXiv preprint arXiv:2501.07301 (2025).
>
> [3] Huang, Hui, et al. "An empirical study of llm-as-a-judge for llm evaluation: Fine-tuned judge model is not a general substitute for gpt-4." Findings of the Association for Computational Linguistics: ACL 2025. 2025.

---

### Official Review · Reviewer_3jyG · 2025-10-31

**Soundness:** 3
**Presentation:** 4
**Contribution:** 3
**Rating:** 8
**Confidence:** 2

**Summary:**

This paper provides a competitor approach to Repeated Sampling for the task of inference time algorithms to improve model performance on complex tasks. Their approach utilizes an exploration phase to sample relevant concepts first and then samples varying responses conditioned on such context to have a larger scope of responses at decision time. The work provides theoretical bounds for how it improves upon RS and additionally presents results on various benchmark complex reasoning datasets demonstrating improvements over baseline methods.

**Strengths:**

1. The problem of scaling inference time compute to generate better performance is interesting and well motivated especially in the context of smaller models.
2. The proposed approach is quite simple and easy to implement, yet very compelling and intuitive to the reader of why this would work well.
3. The theory provides crisp conditions by which GuidedSampling would be preferred to RS.
4. The empirical results are extensive and impressive.

**Weaknesses:**

1. The intuition for Assumption 1 is unclear. That is, if concepts are misapplied or conflicting, the assumption could easily be violated.
2. While the paper well established how candidate solution sets are constructed at inference time, it is still unclear how the final solution is selected from this set. Is it a majority vote?
3. All results are on models of ~3B size. To understand the scale of the solution, their should be more results for various model sizes.

**Questions:**

1. How is the final response selection chosen from the candidate set?
2. How can we ensure that the concepts samples are semantically diverse? If they are not diverse enough, could we ever get worse performance than RS?
3. How sensitive are the results to prompt changes for concept generation?

---

> ### Author Response · Authors · 2025-11-23
>
> We thank the reviewer for finding our work valuable and for their constructive feedback. Below, we address each point and specify the changes we made to the manuscript.
>
> ## Weaknesses
>
> **1.** The intuition behind Assumption 1 is that solving a question becomes easier if we know a good problem‑appropriate “hint” in the form of a concept. Misapplied/conflicting concepts contribute **no requirement** under Assumption 1. Thank you for your comment, and to clarify this further, we have added the following remark for Assumption 1:
>
> We only refer to “relevant” concepts ($c \in C_r$), i.e., concepts that are not misapplied or conflicting with the problem at hand. If a concept is misapplied or conflicts with the problem, it simply lies in $c∉C_r$. Our analysis separates these cases in Eq. (2): the “Recovery from Irrelevant Concepts” term can be small, zero, or occasionally positive (Appendix G provides a concrete case study where an “irrelevant” idea still yields a correct solution).
>
> **2.** In our work, we report pass@k, which can be intuitively understood as checking whether at least one of the $k$ samples is correct (Refer to Section 4 for a formal definition). This doesn’t require us to select a single final solution from the generated candidate solutions. However, it would be interesting to observe the performance of GuidedSampling under a selection mechanism. Thus, during the rebuttal period, we have added a Majority Voting verifier on top of Repeated Sampling, GuidedSampling (best configuration for pass@k; refer to Fig. 5), and Tree-of-Thought. We report these accuracies below:
>
> |Benchmark|Model|Repeated Sampling|GuidedSampling|Tree-of-Thought|
> |-|-|-|-|-|
> |MATH|Llama-3.2-3B-Instruct|50.40|43.40|45.80|
> |GPQA-Diamond|Llama-3.2-3B-Instruct|23.23|23.23|19.19|
> |HumanEval|Llama-3.2-3B-Instruct|20.12|45.12|25.61|
> |OlympiadBench|Llama-3.2-3B-Instruct|17.47|18.35|12.75|
> |MATH|Qwen2.5-3B-Instruct|51.20|64.20|45.40|
> |GPQA-Diamond|Qwen2.5-3B-Instruct|20.71|20.20|7.07|
> |HumanEval|Qwen2.5-3B-Instruct|56.71|50.61|39.02|
> |OlympiadBench|Qwen2.5-3B-Instruct|22.53|21.87|15.27|
> |Average|-|32.80|**35.87**|26.26|
>
> We find that GuidedSampling achieves better accuracy in 4 out of 8 settings, compared to baselines. We have added this finding to Section 5 and the full table in Appendix E. While GuidedSampling performs better than baselines on average, there remains a need for a better selection mechanism, which we leave as future work.
>
> **3.** Our main study uses two 3B open-source LLMs, due to the high computational demands of generating 100 responses and fine-tuning. We have a limited study on GPT-4o-mini (estimated size larger than 3B) and Phi-4-mini-instruct (4B), which also shows improved performance using Guided Sampling (Appendix E). During the rebuttal period, we have included one more model, Gemma-3-27b-it, a 27B model. We showcase the pass@50 performance below with the configuration $K$ = 5 and $M$ = 20 (also included in Appendix E).
>
> | Benchmark | Repeated Sampling | GuidedSampling |
> |-|-|-|
> | MATH | 81.00 | 82.87 |
> |GPQA-Diamond | 70.20 | 91.92 |
> |HumanEval | 83.54 | 94.51 |
>
> We are also working towards adding results for OlympiadBench, as well as ablations for different values of $K$ for all benchmarks. We will update these results in the upcoming days.
>
> ## Questions
>
> **1.** As mentioned in Weakness 2, the goal of the paper is to boost the pass@k performance of models, not select the final solution. Although we conducted a case-study using Majority Voting, building a robust verifier to catch correct solutions (even if in the minority) remains a challenging and future work.
>
> **2.** In the exploration phase, we sample $c_k ⁣∼p_θ(\cdot∣x,c_{1:k−1})$, explicitly conditioning on previous concepts to push the model away from repeats (Section 3.2). The Subsequent Concept prompts in Appendix B.1 require a “new and different” concept and allow early stopping with “No additional concepts found.”. Concepts are stored in a set $C$ (Algorithm 1), eliminating exact duplicates.
>
> However, there are still chances of Repeated Sampling outperforming GuidedSampling when concepts are limited in number (or are similar to each other). We document one such edge case (Section 5): Qwen3.2-3B-Instruct on HumanEval, where it generates only 1.13 distinct concepts on average, resulting in the performance of GuidedSampling dipping below Repeated Sampling (Fig. 4).
>
> **3.** Thank you for your suggestion. In the current work, we haven’t tested GuidedSampling’s sensitivity to prompts for concept generation. We will update the manuscript with these results.

---

> > ### Comment · Reviewer_3jyG · 2025-11-27
> >
> > Thank you for your clarifications. I maintain my positive evaluation of the work.

---

### Official Review · Reviewer_biMm · 2025-11-01

**Soundness:** 2
**Presentation:** 3
**Contribution:** 3
**Rating:** 6
**Confidence:** 3

**Summary:**

This paper proposes GUIDEDSAMPLING, a novel inference-time algorithm designed to address the lack of diversity in traditional Repeated Sampling (RS). GUIDEDSAMPLING resolves this limitation by decoupling the inference process into an exploration phase and a generation phase. It first iteratively generates a diverse set of high-level concepts or theorems in the exploration phase, and then generates candidate solutions for each concept in the generation phase. This method can also be used as a synthetic data generation method. Experimental results show that GUIDEDSAMPLING brings model performance improvements both as an inference-time algorithm and as a synthetic data generation method.

**Strengths:**

1. The core idea of this paper—decoupling the inference process into an exploration phase and a generation phase—is an intuitive, simple, and effective method that directly targets the weakness of RS's lack of diversity.

2. The paper provides a theoretical explanation for when this method works, which is consistent with experimental observations

3. In my opinion, the paper's greater contribution lies in its excellence as a synthetic data generation method. It guides the model to generate more diverse reasoning results on its own, and further allows the model to internalize diversity through training.

**Weaknesses:**

1. The exploration phase is an iterative process, meaning that generating the k-th concept requires waiting for the (k-1) concepts to finish generating. This implies that acquiring K concepts necessitates K serial calls. As an inference-time method, GUIDEDSAMPLING is more challenging to ensure real-time performance compared to RS when computational resources are sufficient.

2. The paper lacks quantifiable comparisons of computational efficiency.

3. The definition of "concepts" appears to be task-specific and manually designed. How can this approach be generalized to more open-ended tasks where concepts are less clearly defined (e.g., commonsense reasoning, long-form story generation)?

4. Although the paper shows that the model can sometimes produce correct solutions even if flawed concepts were generated earlier, a more robust approach would be to implement a feedback loop, allowing the generation phase to inform the exploration phase. This would allow generation efforts to be focused on more promising concepts, rather than wasting compute on flawed paths.

**Questions:**

See weaknesses

---

> ### Author Response · Authors · 2025-11-23
>
> We thank the reviewer for their constructive suggestions. Below, we address each point and specify the changes we made to the manuscript.
>
> ## Weaknesses
>
> **1.** In practice, the value of $K$ is far lower than the number of samples generated (100 in our case). On average, we calculate the pass@50 performance for Repeated Sampling with 100 calls (i.e., for a single instance, we sample 100 solution candidates). We calculated the average number of calls for concept generation across all benchmarks and models, resulting in an average of 4.75 calls per instance for concept generation. Thus, for GuidedSampling, it takes 4.75 (Avg. calls for generating concepts) + 100 (calls for pass@50) = 104.75 calls.  Moreover, as the compute increases (increasing $k$), we observe that the performance gap between Repeated Sampling and GuidedSampling increases in most cases (Fig. 4), suggesting that GuidedSampling benefits more with increased compute. This leads us to believe that when computational resources are sufficient, a small overhead of sequential calls might be a beneficial tradeoff for better performance. We have added this discussion to Section 5.
>
> **2.** We have added a comparison between Repeated Sampling, GuidedSampling, and Tree-of-Thought, showing how pass@50 varies with different compute (number of LLM calls) in Appendix J. We found that GuidedSampling (pass@50=60.2 with 104.75 calls) outperforms both Repeated Sampling (pass@50=48.2 with 100 calls) and Tree-of-Thought (pass@50=37.1 with 154 calls), while being more efficient than Tree-of-Thought. Note that the numbers presented in this answer are averages across all benchmarks and models.
>
> **3.** In our work, we defined “concepts” as the names of the theorems or ideas for domains containing mathematical, scientific, or coding problems. As mentioned in Theorem 1, our framework requires a concept $c$ such that conditioning on $c$ increases the probability of generating the correct solution. In domains such as commonsense reasoning, which involve more imprecise, vague, and uncertain knowledge, defining such concepts is difficult. Hence, the condition stated in Theorem 1, i.e., $P(C_r|x) \gg 0$, might not be satisfied. To verify this, during the rebuttal period, we apply GuidedSampling to Qwen2.5-3B-Instruct on CommonSenseQA, a commonsense benchmark. We asked the model to generate a general idea that could help solve the question (not a task-specific concept, since those are lacking in the commonsense domain). We observed that GuidedSampling underperforms against Repeated Sampling by 3.28% (pass@50) (98.94% RS vs 95.66% GS). Thank you for pointing this out. We acknowledge this as a limitation of GuidedSampling. Based on this case study, we believe that GuidedSampling has a better chance of succeeding when concepts can be formulated efficiently. We have added this study and its corresponding prompts to Appendix E and referenced it in the main results (Section 5) for more insight.
>
> **4.** In the current implementation of GuidedSampling, the Generation phase is far more computationally expensive than the Exploration phase (also discussed in Weakness 1); hence, adding a naive feedback loop and re-running the entire pipeline again based on feedback would increase complexity by a large margin. In our paper, we find that GuidedSampling increases diversity, which leads to better pass@k performance, even at the cost of “irrelevant” concepts (Refer to case study in Appendix G). There may be a possibility that a feedback loop can make GuidedSampling more robust in terms of concept generation; however, this is not in the current scope of the paper. Thank you for suggesting an interesting direction. We have added this in the Future Work Section of the paper.

---

### Official Review · Reviewer_pMb1 · 2025-11-01

**Soundness:** 2
**Presentation:** 2
**Contribution:** 3
**Rating:** 4
**Confidence:** 4

**Summary:**

The paper introduces GuidedSampling, a straightforward and effective method that proposes high-level concepts for the LLM to follow before generating specific solutions. Experiments see performance increase as well as response variety increase with large pass@K sampling.

**Strengths:**

1. By intuition, there is almost no doubt in my mind that the proposed method is effective, as it resembles a mini-agentic approach and directly scales up reasoning through segmentation of hard problems. It's great to see this type of approach being studied in the field, however simple they may be.

2. The empirical justifications are well-discussed, and the results are significant and generalizable.

**Weaknesses:**

1. The theoretical Section 3.3 leaves much to be desired.
    1. Assumption 1 (minor issue: notation $C_{inf}$ should be $C_r$) has four lines of equations, but the second and third are equivalent and both can be derived from the fourth. I suspect the authors wanted to justify the fourth assumption and introduction of $k_c$, but this is not a good way to write assumptions. Furthermore, the fourth line can be relaxed from an equation to a greater-than-inequality, which immediately relaxes the assumption to be much more reasonable, since the original version stipulates two partial probability distributions have the exact same ratio across $x$ and $y^\*$.
    2. Theorem 1 is misleading for using "iff" on Line 281/282: Equation (2) is not equivalent to $P_{GS} > P_{RS}$, but can only derive the latter. This means it is possible that Equation (2) does not hold, but the desired advantage of GS over RS still holds. Despite this, I still think there is value in the theoretical analysis, since the passage under "Sufficient Concept Coverage" is still perfectly valid. I would suggest addressing these problems and rephrasing the theory to be more straightforward and clear.

2. The value of hyperparameter K, the number of concepts, is not clearly specified for the main result in Figure 4.

3. As a motivation, the diversity analysis in Figure 3 is a great way to motivate the method. It would be helpful to do more analyses on diversity of GuidedSampling without finetuning.

**Questions:**

Q1. I'm not sure about the utility of fine-tuning on CCA synthetic data. A naive concatenation of concept + answer may shift the model rollout distribution to exhibit unprecedented behaviors (e.g. become more verbose, suggest unnecessary things), potentially decreasing model generalizability and utiliy. Do models fine-tuned on one synthetic dataset generalize its performance (not diversity) across others? What other validations did you do to ensure no model collapse occurs with CCA fine-tuning?

Q2. Do you have any results on whether the earlier concepts suggested by the concept generator are better than later ones, or vice versa? If there's a pattern there may be a better way to allocate resources than uniformly, but this is not very important so don't stress about it if results are hard to get.

---

> ### Author Response · Authors · 2025-11-23
>
> We thank the reviewer for the careful read and constructive suggestions. Below, we address each point and specify the changes we made to the manuscript.
>
> ## Weaknesses
>
> **1.a.**  Minor Issue: We have updated the manuscript with consistent notation (Section 3.3 and Appendix A).
>
> We state our assumption for the first line, i.e., $I(y;c | x) > 0$. We manipulate it to get the following steps and use the fourth line for Theorem 1. We have condensed the assumption to just line 4 and use the information gain as intuition only. We moved the rest to Appendix A. Finally, we changed equality to a greater-than-inequality. Our revised assumption is as follows:
> *Assumption 1.* There exists a non‑empty set of relevant concepts $C_r$, and a constant $k_c⁡>1$
> such that for any $c \in C_r$
>
> $π_{base}(y^* ∣x,c) ≥ k_c⁡ π_{base}(y^* ∣x)$
>
> **1.b.** Following our assumption, Equation (2) becomes a necessary condition, hence our use of “iff” in Theorem 1. We prove this by contradiction:
>
> Let Equation (2) be false, i.e., Equation (2) is non-positive,
>
> $(k_{min} \cdot P(C_r|x) - 1) \cdot P_{RS}(y^* |x) + \sum_{c \notin C_r} π_{concept}(c|x) \cdot π_{base}(y^* |x, c) \leq 0$
>
> Since the second term is a probability distribution, it can’t be negative. Hence, the first term has to be non-positive, i.e.,
>
> $(k_{min} \cdot P(C_r|x) - 1) \cdot P_{RS}(y^* |x) ≤ 0$
>
> $\Rightarrow k_{min} \cdot P(Cr|x) - 1 ≤ 0$
>
> $\Rightarrow P(C_r|x) ≤ 1 / k_{min}$
>
> $\Rightarrow P(C_r|x) ≤ π_{base}(y^* ∣x) / π_{base}(y^* ∣x,c)$
>
> $\Rightarrow π_{base}(y^* ∣x,c) \cdot P(C_r|x) ≤ π_{base}(y^* ∣x)$
>
> $\Rightarrow P_{GS}( y^* |x) ≤ P_{RS}(y^* |x)$
>
> Therefore, if Equation (2) doesn’t hold, then GuidedSampling underperforms (or is equal) compared to Repeated Sampling. Thus, we believe equation (2) to be a necessary condition in Theorem 1.
>
> **2.** We would like to clarify that all results in Fig. 4 used a total of 100 calls, and we use different values of $K$ for each model and benchmark that maximize pass@50 under this budget. The ablation of various $K$ values vs. pass@50 performance is presented in Fig. 5. We chose the best performing $K$ value corresponding to each benchmark to generate the results presented in Fig. 4. We have referenced Fig. 5 in the caption of Fig. 4 for clarity in the revised manuscript.
>
> In summary, the $K$ value for results in Fig. 4 is the corresponding highest performing $K$ value in Fig. 5.
>
> **3.** We would like to draw your attention to Section 5 under “Diversity in GUIDEDSAMPLING”, where we discuss diversity for base models (without finetuning). For better clarification, we have also included the individual dataset-level diversity values observed in GuidedSampling in the revised Section 5. The values are as follows:
>
>
> | Benchmark | Repeated Sampling | GuidedSampling |
> |-|-|-|
> | MATH | 3.54 | 3.66 |
> | GPQA-Diamond | 6.72 | 7.66 |
> | HumanEval | 2.66 | 3.87 |
> | OlympiadBench | 3.25 | 3.81 |
> | **Average** | 4.04 | 4.75 (17.57% ↑) |
>
> In all benchmarks, we found that GuidedSampling exhibits greater diversity compared to Repeated Sampling, with an average improvement of 17.57% in diversity. We believe that this increase in diversity of candidate solutions is the major factor in improving pass@k values of Repeated Sampling.

---

> ### Author Response · Authors · 2025-11-23
>
> ## Questions
>
> **1.** We would first like to point out that models trained on either FA and CAA trajectories show generalized performance gains across out-of-domain benchmarks, not just diversity. We fine-tuned our models on OpenMathInstruct-2 (a mathematical reasoning dataset) and evaluated on mathematical, scientific, and coding benchmarks (Table 1). Out of the four benchmarks, CAA achieves the best performance on three of them, showing increased generalizability, while FA performs the best on the remaining one. Both of these models outperform the model trained on Repeated Sampling trajectories, showcasing the generalizability of CAA and FA.
>
> Model collapse has been studied to occur for several reasons [1,2]. During the rebuttal, we detect model collapse based on one such observation: a collapsed model can repeat tokens indefinitely without generating an end-of-sequence token during inference. To validate this, during the rebuttal period, we run the base model and the model trained on CAA trajectories on HumanEval with 10 candidate solutions and check the “finish_reason” (https://docs.vllm.ai/en/latest/api/vllm/outputs/#vllm.outputs.CompletionOutput.finish_reason) after generation. Both the base model and the model trained using CAA trajectories return with the finish_reason of “stop”, indicating that the model produced the end-of-sequence token.
>
> Moreover, our evaluation requires the model to generate responses in a particular format (refer to Appendix B.3 for the prompt). Since we did see improvements in pass@k values, we can be assured that the model followed the format closely.
>
> Based on these two insights, we can say with some evidence that no model collapse occurred while training with CAA trajectories. We have added this discussion to Appendix F.
>
> [1] Shumailov, Ilia, et al. "AI models collapse when trained on recursively generated data." Nature 631.8022 (2024): 755-759.
>
> [2] Gerstgrasser, Matthias, et al. "Is model collapse inevitable? breaking the curse of recursion by accumulating real and synthetic data." arXiv preprint arXiv:2404.01413 (2024).
>
> **2.** Thank you for suggesting this ablation. During the rebuttal period, we analyzed all models and benchmarks, which contained a total of 1772 instances. For our analysis, we compute the average accuracy for the $k$-th concept across all questions that produced at least $k$ concepts. We observe a minor decline from $k$=1 to 5 (≈19.8%→16.2%). This observation from the ablation suggests that earlier concepts suggested by the concept generator are better than later ones.
>
> However, another observation for concepts with index $k$ ≥ 6 (i.e., when more exploration is needed) is that they exhibit high performance variance due to a sharp decrease in coverage, e.g., only 72 out of 1772 questions reach $k$ = 9, meaning there are fewer samples with $k$ ≥ 6 concepts. This results in variations in performance, with higher performance being observed for many such cases (e.g., 52.05% performance for $k$ = 14, due to just 23 instances). Thus, although the earlier concepts are beneficial, later ones ($k$ ≥ 6) also contribute to increasing performance, but for a small number of instances that require significant exploration. Hence, for the overall success of GuidedSampling, even the later ones are also important, but the earlier ones play a major role. We have added this discussion to the main results (Section 5).

---

### Author Response · Authors · 2025-11-30

We want to thank all the reviewers for their positive and constructive feedback. Before the reviewer freeze, only reviewer 3jyG acknowledged the rebuttal; therefore, we would like to summarize our efforts during the rebuttal period for other reviewers below:

## Reviewer pMb1

**Weaknesses**

1a. Relaxed the equality of the assumption to greater than or equal to, for a more generalized assumption.

1b. Cleared the confusion regarding the use of "iff" instead of "if" in our theorem.

2. We now explicitly refer to values of $K$ in Fig. 4 for more clarity.

3. Mention the section where we perform diversity analysis and have included individual benchmark-level diversity for more insights.

**Questions**

1. Provided some analysis on why models trained on CAA trajectories might not show model collapse.

2. Performed the ablation suggested by the reviewer and found concept-level pass@k contribution. We found that as the concept index increases, the contribution in pass@k decreases. This gives us the insight *"For the overall success of GuidedSampling, while the later ones are also important, the earlier ones play a major role"*.

## Reviewer biMm

**Weaknesses**

1. Discussed performance gains vs the increased compute required by GuidedSampling.

2. Added performance vs computational charts for all methods used in our study. GuidedSampling outperforms all baselines, while being more efficient compared to baselines like ToT.

3. Explained why GuidedSampling fails to generalize when concepts are not defined clearly, and ran all methods on another OOD benchmark (CommonSenseQA).

4. Discussed why increasing the robustness of concepts is computationally very expensive. Added this discussion to Future Work.

## Reviewer xAGJ

**Weaknesses**

1. Discussed novelty concerns (works already discussed in Related Works), and highlighted our contribution.

2. Cleared up the confusion regarding "irrelevant" concepts under Assumption 1.

3. Explained why GuidedSampling fails to generalize when concepts are not defined clearly, and ran all methods on another OOD benchmark (CommonSenseQA).

**Questions**

1. Discussed why GuidedSampling outperforms ToT, despite being more computationally efficient.


We hope that our rebuttal addresses the concerns raised by the reviewers.

---

### Meta-Review · Area_Chair_nFwm · 2026-01-06

**Summary:**

The paper proposes GuidedSampling as an inference-time algorithm. It tries to resolve the problem of limited sample diversity from Repeated Sampling by incorporating concept exploration into generation. Before the rebuttal, reviewers raised concerns regarding the clarity of assumptions and theorems, the actual computational cost of GuidedSampling, performance on open-ended generation tasks, the need for additional empirical studies, and the difference from prior methods. The authors provided detailed point-by-point responses to these concerns, and the Area Chair believes that the major issues have been largely resolved. As a result, the Area Chair recommends acceptance of this paper as a poster presentation.

**Reviewer Concerns:**

Before the rebuttal, reviewers raised concerns regarding the clarity of assumptions and theorems, the actual computational cost of GuidedSampling, performance on open-ended generation tasks, the need for additional empirical studies, the difference from prior methods, and other minor concerns.

After the rebuttal, the authors improved the clarity of the paper, added additional experiments on response diversity, majority voting, numbers of LLM calls, which resolved most concerns from reviewers.

Some minor concerns remain, such as the difficulty of generating suitable concepts for open-ended generation, which may limit the application scope of the proposed approach.

**Reviewer Scores:**

Reviewer pMb1 would improve their score from 4 to 6 since most of their concerns have been clearly addressed. Other reviewers would maintain their scores.

---

### Decision · Program_Chairs · 2026-01-26

Accept (Poster)